# Unsupervised Anomaly Detection in Tabular Data with Test-time Contrastive Learning

## Abstract

Unsupervised anomaly detection methods typically learn the feature patterns of normal samples during training, subsequently identifying samples that deviate from the learned patterns as anomalies during testing. However, most existing methods assume that the normal patterns in the test set are similar to those in the training set, ignoring the fact that a limited number of training samples may not cover all possible normal patterns. As a result, when the normal patterns in the test set differ from those in the training set, the model may struggle to distinguish whether these samples are normal or anomalous, leading to incorrect predictions. To address this issue, we propose a novel **T**est-time **C**ontrastive learning approach for unsupervised **A**nomaly **D**etection in tabular data (namely TCAD). Specifically, TCAD consists of two core stages: Collaborative Dual-task Training and Test-Time Contrastive Learning. In training, Collaborative Dual-task Training uses two self-supervised tasks to capture multi-level features of normal samples and model normal patterns. At test time, Test-Time Contrastive Learning assigns pseudo labels to high-confidence samples and updates the model in two ways: First, it facilitates model adaptation to pseudo-normal samples while preventing overfitting to pseudo-abnormal ones. Second, it employs a KNN-based contrastive strategy to align pseudo-normal samples with the training distribution while pushing pseudo-abnormal samples away. By combining robust normal pattern modeling with iterative test-time adaptation, TCAD improves anomaly discrimination, especially under distribution shifts between training and test sets. We construct distribution shifts on 15 widely used tabular datasets, and the results show that TCAD achieves state-of-the-art performance, outperforming the best baseline by 4.19% in AUC-ROC, 3.15% in AUC-PR, and 6.64% in F1 score.

## 1 Introduction

Anomaly detection aims to identify data points that deviate significantly from the majority of instances in a dataset (Zha et al., 2020) and plays a crucial role in various fields, such as medical diagnosis (Fernando et al., 2021), network intrusion detection (Ahmad et al., 2021; Qiao & Pang, 2023; Qiao et al., 2024b; Niu et al., 2024), financial fraud detection (Al-Hashedi & Magalingam, 2021; Qiao et al., 2024a; 2025), and industrial inspection (Liu et al., 2024). Due to the difficulty of obtaining labeled anomaly data in real-world scenarios, unsupervised anomaly detection methods that utilize only normal samples for training have become the mainstream approach.

Existing unsupervised anomaly detection methods can be broadly classified into 4 categories: one-class classification methods (Schölkopf et al., 1999; Ruff et al., 2018), clustering/feature-distribution-based methods (Liu et al., 2022; Ali et al., 2024; Li et al., 2022), reconstruction-based methods (Schlegl et al., 2017; Gong et al., 2019), and self-supervised learning methods (Schlegl et al., 2017; Gong et al., 2019). Although these methods differ in their design strategies, they share a common core idea: learning feature patterns from normal samples during the training phase and identifying test samples that deviate from these learned patterns as anomalies during the testing phase.

However, most existing methods overlook the fact that limited training samples cannot encompass all possible patterns of normal samples. When the representations of normal test samples deviate from the learned representation space, it becomes challenging for the model to distinguish between

normal and abnormal samples, resulting in many incorrect predictions. As illustrated in Fig. 1, case (a) represents a scenario where the representations of normal samples in the test set are aligned with those in the training set, maintaining a clear separation from the representations of abnormal samples. In contrast, case (b) depicts a scenario where the representations of normal samples in the test set are misaligned with those in the training set. Existing methods typically identify anomalies by detecting samples that deviate from known representation space during testing. Therefore, in case (b), they classify test samples that are misaligned with the representations of normal samples in the training set as anomalies, resulting in many incorrect classifications. Intuitively, allowing a trained model to adapt to the test data could help align the representation spaces of the training and test sets, thereby reducing the likelihood of falsely identifying normal test samples as anomalies. However, this adaptation poses a critical challenge: if the model learns representations of potentially anomalous samples during the adaptation process, it may compromise its ability to distinguish between normal and abnormal instances, ultimately impairing anomaly detection. Therefore, it is crucial to ensure that the model retains its discriminative capability while undergoing adaptive optimization.

In this paper, we propose TCAD, a novel test-time contrastive learning approach for unsupervised anomaly detection in tabular data. TCAD comprises two key stages: (1) Collaborative Dual-task Training, and (2) Test-Time Contrastive Learning. During training, TCAD employs a dual-task learning framework that integrates a main task and an auxiliary task to capture multi-level feature representations of samples. This design facilitates the effective extraction of latent patterns associated with normal samples in the training set. During testing, instead of indiscriminately assigning high anomaly scores to all samples that deviate from known normal patterns, TCAD employs test-time contrastive learning (TTCL) to refine the anomaly detection process. As shown in Fig. 1(c), at test time, the core idea of TTCL is to enable the model to learn multi-level representations of test samples, while optimizing their embeddings such that pseudo-normal samples are pulled closer to the training distribution (blue dashed circle) and pseudo-abnormal samples are pushed away from it (red dashed circle). Specifically, TTCL first assigns pseudo-labels to high-confidence samples predicted by the model. It then employs self-supervised tasks to help the model learn multi-level feature representations of pseudo-normal samples, while avoiding accurate reconstruction of pseudo-abnormal samples. Next, TCAD performs k-nearest neighbor (KNN) contrastive learning in the embedding space to optimize the feature representations of samples. Finally, the

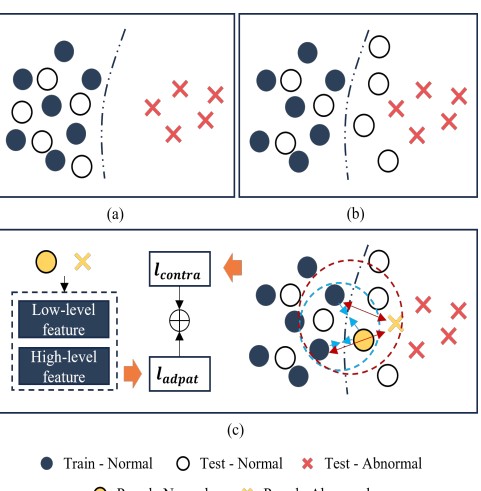

Figure 1: Two cases of the sample distribution. (a) The normal pattern in the test set aligns with that in the training set, while remaining far from the abnormal pattern. (b) The normal pattern in the test set is misaligned with that in the training set and overlaps with the abnormal pattern. (c) During the test-time contrastive learning, TCAD employs self-supervised tasks to learn multi-level features of pseudo-normal samples, while avoiding accurate reconstruction of pseudo-abnormal samples. Additionally, TCAD employs k-nearest neighbor contrastive learning to pull pseudo-normal samples closer to the training distribution and push pseudo-anomalous samples away from it.

optimized model is used to repeat the same process on the remaining unlabeled test samples until all samples are assigned pseudo-labels. Extensive experiments on 15 tabular datasets demonstrate that our method achieves state-of-the-art performance, outperforming the previous best approach by nearly 7% in average F1 score across all datasets.

Our main contributions can be summarized as follows:

- We investigate a practical yet underexplored problem in unsupervised anomaly detection, where the distribution of normal samples in the test set deviates from that in the training set.

- We propose a novel test-time contrastive learning approach for unsupervised anomaly detection in tabular data under the studied setting, which enhances the model's ability to effectively distinguish test samples that deviate from known normal patterns.

- Two core stages, Collaborative Dual-task Training and Test-Time Contrastive Learning, are designed to capture multi-level feature representations of samples and enhance the model's capacity to discriminate between normal and abnormal samples during testing.

- Extensive experiments on 15 tabular datasets demonstrate that our method outperforms 13 unsupervised anomaly detection baseline models. TCAD outperforms the best baseline by 4.19% in AUC-ROC, 3.15% in AUC-PR, and 6.64% in F1 score.

## 2 RELATED WORK

### 2.1 UNSUPERVISED ANOMALY DETECTION

Unsupervised anomaly detection, which does not rely on anomaly labels during the training phase, is one of the most practical approaches to anomaly detection. Existing studies typically aim to learn the underlying feature patterns of normal samples during the training phase by modeling their feature distributions, densities, compact embeddings, and internal structures. At test time, samples that significantly deviate from these learned normal patterns are classified as anomalies. These methods can be broadly classified into 4 categories: One-class classification-based methods (Schölkopf et al., 1999; Tax & Duin, 2004; Ruff et al., 2018; Goyal et al., 2020; Massoli et al., 2021; Xu et al., 2024) learn a decision boundary that encloses the normal samples, classifying those that fall outside this boundary as anomalies during testing. Clustering/feature-distribution-based methods (Breunig et al., 2000; Zong et al., 2018; Liu et al., 2022; Ali et al., 2024; Li et al., 2022) detect anomalies by estimating the density of data points or evaluating their positions within the feature distribution. Reconstruction-based methods (Schlegl et al., 2017; 2019; Gong et al., 2019; Zavrtanik et al., 2021; Zaheer et al., 2022; Zhang et al., 2023; Guo et al., 2024) learn compact embeddings to model normal feature patterns and classify samples with high reconstruction errors as anomalies. Self-supervised learning-based methods (Bergman & Hoshen, 2020; Qiu et al., 2021; Shenkar & Wolf, 2022; Yin et al., 2024) design auxiliary tasks to uncover latent data structures and patterns; samples that fail these tasks at test time are flagged as anomalies.

Although these methods adopt a variety of model architectures, they all share the assumption that the distribution of normal samples in the test set is similar to that in the training set. As a result, all samples that deviate from the training distribution are detected as anomalies. In contrast, our proposed method explicitly addresses the practical challenge where the distribution of normal samples in the test set may differ from that in the training set. Consequently, it achieves improved detection performance on datasets exhibiting distributional shifts.

### 2.2 TEST-TIME ADAPTION

Test-time training (TTT), a domain adaptation approach, aims to mitigate performance degradation caused by domain shift by enabling models to adapt to test data without labels. It introduces a self-supervised task during training, which is later used to fine-tune the model at test time, improving robustness to distributional changes (Kouw & Loog, 2018; Liu et al., 2021a; Sun et al., 2020). Various self-supervised tasks have been proposed to enhance model performance during the testing phase, including rotation prediction(Feng et al., 2019; Sun et al., 2020), moment matching(Long et al., 2018; Liu et al., 2021b), entropy minimization(Shu et al., 2022), self-training (Zou et al., 2019; Jang et al., 2023), etc. In addition, several approaches have been proposed recently in anomaly detection to address the issue of distribution shift. For example, AnoShift (Dragoi et al., 2022) constructs datasets where shifts naturally emerge over time. Kim et al. (2024) adapt models at test time to potential normal data in time series anomaly detection tasks. OWAD (Han et al., 2023) combines human supervision with unsupervised methods to reduce the labeling cost induced by distribution shift. Cao et al. (2023) and Carvalho et al. (2023) tackle image anomaly detection by learning distribution-invariant representations to mitigate the shift problem.

Although TTT has achieved promising results in many tasks, its application to unsupervised anomaly detection tasks remains challenging. The fundamental reason lies in the fact that unsupervised

anomaly detection models are trained without access to anomalous samples, while such anomalies may exist in the test set. If the model inadvertently learns the feature representations of anomalies during adaptation, it may entirely lose its ability to detect them. To address this issue, our proposed method explicitly considers the possible presence of anomalies during test-time training. It prevents the model from accurately modeling abnormal patterns and pushes anomalous samples away from the learned distribution.

## 3 METHODOLOGY

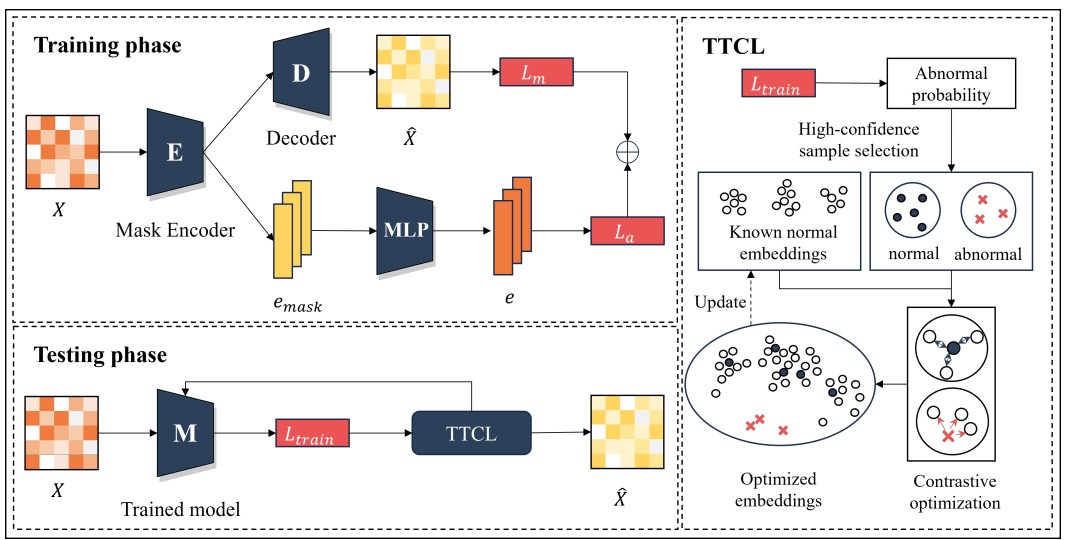

Figure 2: The overview of TCAD. During the training phase, TCAD utilizes two reconstruction tasks to extract multi-level feature information from the training samples. The main task reconstructs the input samples to capture low-level features, while the auxiliary task reconstructs their embeddings to capture high-level features. During the testing phase, TCAD iteratively updates the trained model using Test-Time Contrastive Learning, which adapts the model to the characteristics of high-confidence samples while refining their embedding distributions relative to the training samples.

### 3.1 PROBLEM STATEMENT

In this paper, we focus on unsupervised anomaly detection methods in tabular data that do not rely on labeled anomalies during training. Specifically, these methods are trained solely on datasets composed of normal samples. Given the training set $\mathcal{D}_{train} = \{\mathbf{x}_i^{train}\}_{i=1}^{N}$ and test set $\mathcal{D}_{test} = \{\mathbf{x}_i^{test}\}_{i=1}^{N'}$, $N$ and $N'$ represent the number of training set and the test set respectively, $\alpha$ is the contamination rate of the test set $\mathcal{D}_{test}$. Unsupervised anomaly detection model $M$ is trained on $\mathcal{D}_{train}$ to learn the feature pattern of the normal samples, then the trained model $M_{train}$ is employed to predict the anomaly probabilities of test samples $P_{abnormal}$. Samples with high anomaly probabilities are detected as anomalies $\mathcal{A}$. The process of model training and label predicting can be formulated as $M_{train} = M(\mathcal{D}_{train})$, $P_{abnormal} = M_{train}(\mathcal{D}_{test})$, $\mathcal{A} = \mathcal{D}_{test}(Norm(P_{abnormal}) > 1 - \alpha)$, where $Norm$ is the Min-Max Scaler, which scales the anomaly probability $P_{abnormal}$ to $[0, 1]$.

### 3.2 OVERVIEW OF THE PROPOSED TCAD

The key insight of TCAD lies in enabling the model to adapt to normal test samples whose distribution differs from that of the training set, while mitigating the risk of learning abnormal features that could degrade anomaly detection performance. To achieve this, TCAD employs two core stages: Collaborative Dual-task Training and Test-Time Contrastive Learning, which are responsible for training and test-time adaptation, respectively. As shown in Fig. 2, TCAD captures both low-level and high-level features of tabular data during the training phase through a main task and an auxiliary task, effectively learning the normal patterns. During the testing phase, TCAD performs Test-Time

Contrastive Learning to refine the trained model. Specifically, TTCL first assigns pseudo labels to high-confidence outputs of the trained model and then leverages the two training-phase tasks to learn the feature representations of pseudo-normal samples while preventing the model from accurately learning pseudo-abnormal patterns. Second, a KNN-based contrastive learning module is designed to pull pseudo-normal samples closer to the training distribution and push pseudo-abnormal samples away from it. Through this process, the model gradually improves its ability to distinguish between normal and abnormal test samples that deviate from the training distribution using the high-confidence samples selected in each iteration. Finally, the optimized model is repeatedly fine-tuned on the remaining unlabeled samples using the same procedure until pseudo labels are assigned to all test samples.

## 3.3 COLLABORATIVE DUAL-TASK TRAINING

The model's ability to capture normal patterns is closely tied to the richness of informative features extracted from normal samples during training. To strengthen this capability, TCAD employs a collaborative dual-task learning method that integrates two complementary tasks (main task and auxiliary task), enabling the model to learn multi-level feature representations of samples effectively.

**Model Details.** The backbone of the model is built upon a masked autoencoder. Give the input $\mathbf{X} \in \mathbb{R}^{B \times d}$ from the training set $\mathcal{D}_{train}$, $B$ is the batch size, $d$ denotes the dimension of the feature vector. The input $\mathbf{X}$ is first passed through the masked encoder $E$, which serves as a shared feature extractor for both tasks. This mask encoder $E$ consists of two components: a mask generator $g_1$ and an encoder $g_2$, $E = g_1 + g_2$. $g_1$ produces multiple mask tensors $\mathbf{X}_{mask} = g_1(\mathbf{X})$ of the same size as $\mathbf{X}$, and leverage a sigmoid function to scale each value of $\mathbf{X}_{mask}$ between 0 and 1. Element-wise multiplication is then applied between $\mathbf{X}_{mask}$ and $\mathbf{X}$, the obtained masked input is subsequently passed into $g_2$ to obtain the masked representation $\mathbf{e}_{mask} = E(\mathbf{X}) = g_2(\mathbf{X}_{mask} \odot \mathbf{X})$ in the embedding space. Furthermore, to capture a broader spectrum of information from normal samples, we ensure sufficient diversity in the masking patterns. This is essential, as using similar masks may cause the model to learn redundant features, which not only fail to improve anomaly detection performance but may also degrade it. Inspired by MCM (Yin et al., 2024), the diversity of masking patterns is promoted by incorporating a dedicated loss function, as defined in $\mathcal{L}_{\text{div}} = \sum_{i=1}^{T} \left[ \ln \left( \sum_{j=1}^{T} \left( \mathbb{I}_{i \neq j} \cdot e^{\frac{<\mathbf{x}_{mask}^i, \mathbf{x}_{mask}^j>}{\tau}} \right) \right) \cdot s \right]$, where $<>$ denotes the inner product operation, $\mathbb{I}_{i \neq j}$ is the indicator function, if $i = j$, $\mathbb{I}_{i \neq j} = 0$, otherwise $\mathbb{I}_{i \neq j} = 1$, $\tau$ is a temperature parameter, and $s$ is a scaling factor to adjust the range of the diversity loss, $T$ denotes the number of masks.

**Main task: learning low-level features.** In the main task, the masked representation is fed into the decoder $D$ to reconstruct the original input $\mathbf{X}$, as shown in $\hat{\mathbf{X}} = D(\mathbf{e}_{mask})$. By minimizing the reconstruction loss $\mathcal{L}_m = \frac{1}{T} \sum_{i=1}^{T} \|\hat{\mathbf{X}}_i - \mathbf{X}\|^2$ between the input and its reconstruction, the model learns low-level feature representations of the tabular data.

**Auxiliary task: capturing high-level features.** In the auxiliary task, the masked representation $\mathbf{e}_{mask}$ is fed into a multi-layer perceptron (MLP) to reconstruct the embedding $\mathbf{e}$ of the unmasked input. By minimizing the reconstruction loss between the predicted and original embeddings, the model captures the intrinsic knowledge embedded in the encoded representations, thereby learning high-level feature representations of the data. To ensure that $\mathbf{e}$ and $\mathbf{e}_{mask}$ have the same size, we replicate $\mathbf{X}$ $T$ times to match the size of $\mathbf{X}_{mask}$, and then pass the replicated input through the encoder $g_2$ to obtain $\mathbf{e} = g_2(\mathbf{X}^T)$, $\mathbf{X}^T$ represents the input $\mathbf{X}$ that has been replicated $T$ times. The auxiliary task is trained by minimizing the reconstruction loss $\mathcal{L}_a = \frac{1}{T} \sum_{i=1}^{T} \|\hat{\mathbf{e}}_i - \mathbf{e}\|^2$ between the predicted embedding $\hat{\mathbf{e}} = MLP(\mathbf{e}_{mask})$ and the embedding $\mathbf{e}$.

**Model Training Loss.** The overall training loss of the model integrates the reconstruction losses from the main and auxiliary tasks, as well as the mask diversity loss, and is formally defined as $\mathcal{L}_{Train} = \mathcal{L}_m + \lambda \mathcal{L}_a + \gamma \mathcal{L}_{div}$, where $\lambda$ and $\gamma$ are the weights used to adjust the overall loss function.

## 3.4 TEST-TIME CONTRASTIVE LEARNING

During the testing phase, prior unsupervised anomaly detection methods typically apply the trained model directly to estimate anomaly scores for test samples, without accounting for the possibility

that the distribution of test data may differ from that of the training data. This oversight hampers the model's adaptability on certain datasets, thereby limiting its detection accuracy. To address this issue, we propose a Test-Time Contrastive Learning approach to update the trained model during the testing phase.

**High-Confidence Samples Selection.** Given the test set $\mathcal{D}_{test}$, we first apply the trained model to output the losses for all test samples and normalize them into the range $[0, 1]$. Then the normalized losses of test samples can be regarded as their anomaly probability, $P_{abnormal} = Norm(M_{train}(\mathcal{D}_{test}))$, where $M_{train}$ represents the trained model, $\mathcal{P}_{abnormal}$ denotes the anomaly probabilities of all test samples, $Norm$ denotes the Min-Max Scaler. Subsequently, TTCL selects the most confident normal and abnormal samples from the test set based on sorted anomaly scores, referring to them as pseudo-normal and pseudo-abnormal samples. The confidence threshold for sample selection is manually specified. Its default value is 10%, with a predefined normal–abnormal ratio of 5:1. In addition, similar to prior works (Ruff et al., 2018; Li et al., 2022; Yin et al., 2024) that assumes access to the true contamination rate, our method can also use the actual contamination rate as threshold whenever it is available. The selected samples can be denoted as $\mathcal{H}_{normal} = \{h_i^{normal}\}_{i=1}^{C_{normal}}, \mathcal{H}_{abnormal} = \{h_i^{abnormal}\}_{i=1}^{C_{abnormal}}$, where $\mathcal{H}_{normal}$ represents the set of high-confidence normal samples and $\mathcal{H}_{abnormal}$ represents the set of high-confidence abnormal samples, $C_{normal}$ and $C_{abnormal}$ denote the number of samples in two sets.

**Model Adaptation to Selected Samples.** At the test time, TTCL leverages both the main and auxiliary tasks to adapt to feature representations of the selected samples. Since these two tasks are trained without requiring labels, the approach does not pose any risk of test label leakage. Specifically, the model adapts to pseudo-normal and pseudo-abnormal samples separately. The goal of adapting to pseudo-normal samples is to learn their feature representations and reduce their reconstruction errors, preventing them from being mistakenly identified as anomalies. Conversely, the adaptation to pseudo-abnormal samples aims to hinder the model from accurately learning their representations, so that they yield high errors during inference and are correctly identified as anomalies. The loss function of the model adaptation is shown in eq. (1).

$$\mathcal{L}_{adapt} = \sigma_s \cdot \frac{1}{C_s} \sum_{i=1}^{C_s} \left( \mathcal{L}_m(h_i^s) + \lambda \mathcal{L}_a(h_i^s) + \gamma \mathcal{L}_{div} \right), \ \sigma_s = \begin{cases} +1, & s = \text{normal} \\ -1, & s = \text{abnormal} \end{cases} \tag{1}$$

**Embedding Contrastive Optimization.** In addition to adapting the model to the feature representations of the selected samples, TTCL further optimizes their representations in the embedding space. Specifically, TTCL first maps the selected samples into the embedding space of the training data using the trained model, and then encourages pseudo-normal samples to move closer to the training distribution while pushing pseudo-anomalous samples away from it. However, requiring each high-confidence test sample to be uniformly close to or distant from all training samples is both unrealistic and inefficient. This is because normal samples exhibit diverse patterns, and a test sample is unlikely to be close to all modes present in the training data. As a result, it is naturally distant from some training samples. Moreover, computing distances to all training samples incurs prohibitive computational costs. Therefore, TTCL adopts a KNN-based contrastive learning strategy that only utilizes the k nearest neighbors of selected samples to either pull pseudo-normal samples closer to the training distribution or push pseudo-abnormal samples further away from it. This localized contrastive approach effectively refines the embedding positions of selected samples, enhancing the model's discriminative power while improving optimization efficiency. The contrastive loss function is defined as eq. (2).

$$\mathcal{L}_{contra} = \sigma_s \cdot \frac{1}{C_s} \sum_{i=1}^{C_s} \|h_i^s - KNN(h_i^s, \mathcal{O}, k)\|^2, \ \sigma_s = \begin{cases} +1, & s = \text{normal} \\ -1, & s = \text{abnormal} \end{cases} \tag{2}$$

where $\mathcal{O}$ denotes the embeddings of known normal samples, $KNN(\mathbf{x}, \mathcal{O}, k)$ denotes finding the $k$-nearest embeddings to the embedding representation of sample $\mathbf{x}$ from the set of known normal embeddings $\mathcal{O}$.

**Model Update Loss.** For pseudo-normal or pseudo-anomalous samples, the model jointly optimizes the adaptation loss and the contrastive loss during the update process. The overall loss function is defined as $\mathcal{L}_{Update} = \mathcal{L}_{adpat} + \mathcal{L}_{contra}$, where $\delta$ is a hyperparameter to balance two losses.

**Update Iterations.** Let the originally trained model $M_{\text{train}}$ be denoted as $M_{update}^{(0)}$, and let $n$ denote the total number of update rounds. During test time, the model is iteratively refined using TTCL. In each round, $M_{update}^{(n-1)}$ is updated using the current pool of known normal samples, $M_{update}^{(n)} = TTCL(M_{update}^{(n-1)}; \mathcal{L}_{Update})$. Subsequently, newly identified high-confidence normal samples are added to the pool for the next round, $\mathcal{O}^{(n)} = \mathcal{O}^{(n-1)} + \mathcal{H}_{normal}^{(n-1)}$. This process is repeated iteratively until the remaining unselected samples are insufficient for further selection. Finally, the updated model are used to predict the labels $y^{test}$ of the test samples, as illustrated in $p^{test} = M_{update}^{(n)}(\mathcal{D}_{test})$, $y_i^{test} = \mathbb{I}(p_i^{test} \geq Percentile(p^{test}, \alpha)$, where $p^{test}$ denotes the predicted anomaly probabilities of test samples, $y_i^{test}$ denotes the predicted label of test sample $i$, $\mathbb{I}(\cdot)$ denotes 1 if the condition $\cdot$ is met, and 0 otherwise, $Percentile(p^{test}, \alpha)$ represents the value at the $100 * \alpha\%$ percentile in $p^{test}$.

# 4 EXPERIMENTS

## 4.1 DATA SHIFT CONSTRUCTION AND ANALYSIS

Following prior works (Li et al., 2022; Shenkar & Wolf, 2022; Yin et al., 2024), we first select 15 commonly used tabular datasets from ODDS (Rayana, 2016) and ADBench (Han et al., 2022), covering a wide range of domains, scales and feature dimensions. These diverse datasets enhance the generality of our evaluation and strengthens the reliability of the conclusions. Detailed statistics of datasets are provided in Appendix A.2.

Second, we apply K-Means clustering to all normal samples in each dataset. The majority of samples from the largest cluster are used as the training set, while the remaining samples from this cluster, together with the samples from the other clusters and the anomalous samples, form the test set. In this way, the normal samples in the test set consist partly of data consistent with the training distribution and partly of data deviating from it. To verify the existence of such shifts, we follow the distribution shift protocol of AnoShift (Dragoi et al., 2022) and examine the processed datasets using t-SNE visualization, Jeffreys Divergence (JD) and the Optimal Transport Dataset Distance (OTDD). JD is computed feature-wise via normalized histograms and sums the forward and reverse KL divergences, thus reflecting probability differences across features. OTDD, in contrast, is obtained by solving an optimal transport problem in the original feature space, capturing the geometric discrepancy between datasets. In our experiments, OTDD values are normalized to the range [0, 1].

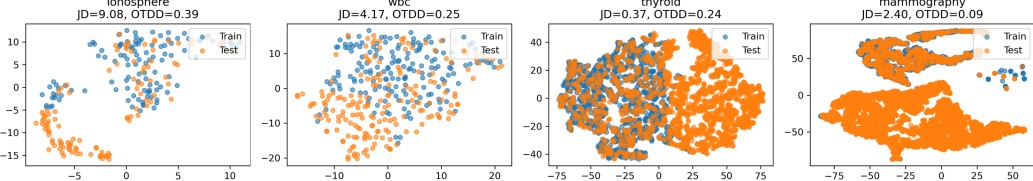

Figure 3: A comparison of the normal distributions in the training and test sets visualized using t-SNE.

As shown in Figure 3, the distribution of normal samples in the test set (orange points) exhibits a clear shift from that in the training set (blue points). The high values of JD and OTDD further corroborate this observation. Therefore, the constructed distribution shift is indeed substantial and well grounded. The visualizations of samples, feature distributions, as well as the detailed values of JD and OTDD for all datasets are provided in Appendix A.3.

## 4.2 EXPERIMENTAL SETUP

**Competing methods.** We compare TCAD against 13 prominent baseline methods to demonstrate its effectiveness. They can broadly be divided into five categories: one-classification-based methods (OCSVM (Schölkopf et al., 1999), DeepSVDD (Ruff et al., 2018)), neighbor-based/feature-distribution-based methods (LOF (Breunig et al., 2000), Iforest (Liu et al., 2008), DIF (Xu et al., 2023a), ECOD (Li et al., 2022), LUNAR (Goodge et al., 2022)), reconstruction-based methods

(MCM (Yin et al., 2024)), self-supervised learning-based methods (GOAD (Bergman & Hoshen, 2020), NeuTral AD (Qiu et al., 2021), ICL (Shenkar & Wolf, 2022), SLAD (Xu et al., 2023b)), and representation learning-based method(DRL (Ye et al., 2025)). Detailed descriptions of all methods are provided in Appendix A.5.

**Evaluation Metrics.** Following the previous study (Shenkar & Wolf, 2022; Yin et al., 2024; Ye et al., 2025), we employ Area Under the Precision-Recall Curve (AUC-PR), Area Under the Receiver Operating Characteristic Curve (AUC-ROC) and F1 score as our evaluation criteria.

**Implementation details.** All experiments are conducted on NVIDIA GeForce RTX 2080 Ti with PyTorch (Paszke et al., 2019). During the training phase, the epochs are set to 200, the batch size is 512, the optimizer is Adam, the weight decay is 1e-5, the scheduler is ExponentialLR, and the gamma is 0.98. During the test phase, the value $k$ is set to 3 for all data sets. The hyperparameter $\lambda$ is set to $\min(1.0, 1.0/\mathcal{L}_m)$, which enables adaptive adjustment of the weights between the main task and the auxiliary task across different datasets. The value of $\gamma$ is set following the configuration used in MCM (Yin et al., 2024), and $\delta$ is set to 1 by default. IForest, LOF, OCSVM, DeepSVDD, ECOD and LUNAR are implemented by the Pyod library (Zhao et al., 2019). DIF, GOAD, NeuTralAD, ICL and SLAD are implemented by DeepOD library (Xu et al., 2023a; 2024). MCM is implemented based on their official open-source code releases. All results of the main experiments and ablation experiments are calculated by averaging the results from the three independent training runs. The results of other experiments are obtained by training the model with a single run.

### 4.3 EMPIRICAL RESULTS AND ANALYSIS

**Main Results.** We visualize the evaluation results of all methods across all datasets using box plots , and we additionally provide confidence intervals for the main metrics. The detailed procedure for computing the confidence intervals is provided in Appendix A.4. Figure 4 summarizes the numerical results of AUC-PR and AUC-ROC obtained from evaluating 14 methods across 15 datasets. Subfigures (a) and (b) illustrate the distributions of AUC-PR and AUC-ROC values across datasets, respectively, while subfigures (c) and (d) present the corresponding distributions of rankings for AUC-PR and AUC-ROC. According to the results in the figure, TCAD achieves the best average performance across datasets in terms of both AUC-PR and AUC-ROC. Specifically, TCAD outperforms the best baseline by 3.15% in AUC-PR, with an average ranking advantage of 1.27 positions, and surpasses the best baseline by 4.19% in AUC-ROC, with an average ranking advantage of 0.94 positions.

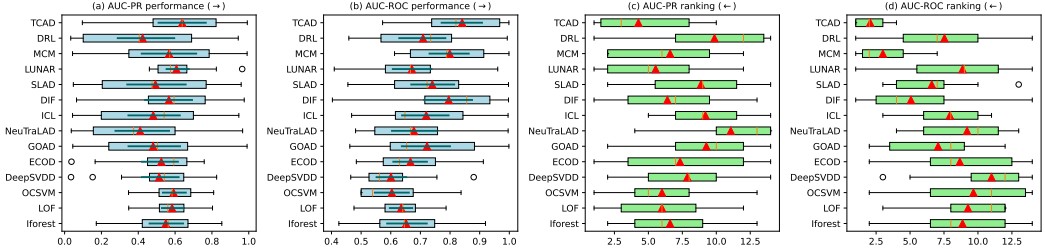

Figure 4: Comparison of all models' performance and ranking across different datasets in terms of AUC-PR and AUC-ROC. The triangles represent the average value over all datasets.

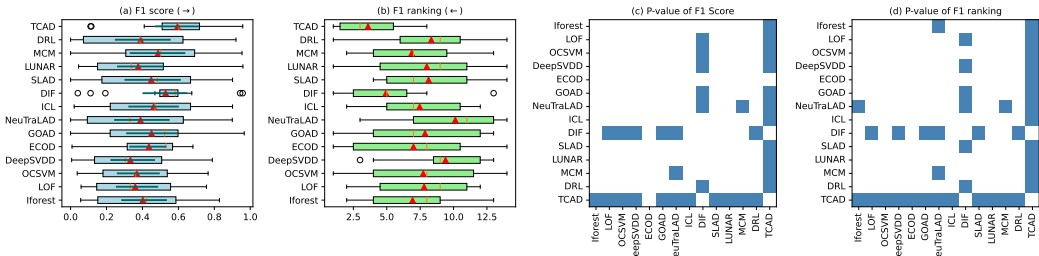

Figure 5: Comparison of F1 scores. (a) and (b) compare the F1 scores and rankings of all models across different datasets, where the triangles denote the averages over all datasets. (c) and (d) conduct Wilcoxon tests across models and datasets. Blue cells indicate corresponding p-values below 0.05 (significant), while white cells indicate p-values above 0.05 (not significant).

Figure 5 presents the F1 performance of different methods. Specifically, subfigures (a) and (b) illustrate the distribution of F1 scores across all datasets and the corresponding ranking distributions, respectively. TCAD achieves an average F1 score that surpasses the second-best model by 6.64%, and an average ranking advantage of 1.33 positions. Subfigures (c) and (d) employ the Wilcoxon signed-rank test (Woolson, 2007) (with $\alpha = 0.05$) to assess the statistical significance of the improvements. At the 95% confidence level, the improvements of TCAD over the baseline models are statistically significant on the majority of datasets. The above results demonstrate the effectiveness of TCAD in detecting anomalies under distribution shift. Detailed results on all datasets are available in the Appendix A.6.

**Results on the AnoShift Subsets.** Beyond the main experiments, we further investigate distributional shifts induced by temporal evolution and conduct additional evaluations on subsets of the AnoShift benchmark. Specifically, due to computational resource constraints, we use the

Table 1: AUC-ROC comparison between MCM and TCAD on the Anoshift subsets.

| Method/Year | 2011 | 2012 | 2013 | 2014 | 2015 |
|---|---|---|---|---|---|
| MCM | 0.9445 | 0.8381 | 0.8341 | 0.3620 | 0.2963 |
| TCAD | 0.8962 | 0.9038 | 0.8692 | 0.4735 | 0.3778 |

2006–2010 valid sets as the training data and the 2011–2015 valid sets as the test data. Since each valid set follows the same distribution as its corresponding full-year dataset, this setup effectively simulates temporal distribution shifts on a smaller scale.

Following the AnoShift evaluation protocol, AUC-ROC is employed to evaluate performance of method. We select MCM with the highest AUC-ROC in baseline models as the compaison model, and the experimental results are presented in Table 1. MCM only outperforms TCAD on the 2011 test set, which is closest to the training years. As the distribution shift becomes more pronounced over time, TCAD consistently surpasses MCM on the 2012–2015 test sets. The largest improvement is observed in 2014, where TCAD exceeds MCM by 0.1115. These results demonstrate that TCAD maintains strong and competitive detection performance under distribution shifts induced by temporal evolution.

**Analysis of Pseudo-Label Noise Impact.** In the TTCL module of TCAD, pseudo-labels are assigned to samples with high-confidence prediction probabilities. To investigate whether noisy pseudo-labels continuously impair model performance, we select four datasets with notably different overall performance (i.e., pendigits, cardiotocography, car-

Table 2: True rate of pseudo labels in early iterations (true rate of normal labels-true rate of abnormal labels).

| Dataset | iter 1 | iter 2 | iter 3 | iter 4 |
|---|---|---|---|---|
| pendigits | 1.00-0.07 | 1.00-0.00 | 1.00-0.00 | 1.00-0.00 |
| cardiotocography | 0.98-0.59 | 0.75-0.52 | 0.96-0.24 | 0.74-0.67 |
| cardio | 1.00-0.88 | 1.00-0.82 | 1.00-0.76 | 0.91-0.88 |
| breastw | 1.00-1.00 | 1.00-1.00 | 1.00-1.00 | 1.00-1.00 |

dio, and breastw) and track their label accuracy during early iterations to examine whether errors are persistently amplified. The statistical results are reported in Table 2, the accuracy of the pseudo labels fluctuates rather than continuously declining. In addition, even on the pendigits dataset, where the model performs the worst and the true anomaly rate eventually drops to zero, the final detection performance of TCAD still surpasses that of most baselines. These phenomenons demonstrates that: (1) Label noise does not cause persistent degradation in model performance. (2) Despite the presence of noisy labels, the benefits gained from utilizing them outweigh their potential drawbacks.

Furthermore, we introduce a co-teaching mechanism in which two lightweight MLPs mutually select low-loss samples to reduce noise in pseudo-labels. We experiment with forget rates of 10%, 20%, 30%, and 40%, and report the results in Table 3. Interestingly, increasing the forget rate does not consistently yield better performance, likely because the additional supervision introduced by the co-teaching models may itself introduce errors and misjudge some pseudo-labels. Nevertheless, with a 40%

Table 3: Average detection performance across 15 datasets under different forget rates for filtering pseudo-label noise.

| Forget rate | auc-roc | auc-pr | pr |
|---|---|---|---|
| 0% | 0.8408 | 0.6388 | 0.5953 |
| 10% | 0.7710 | 0.5829 | 0.5351 |
| 20% | 0.8201 | 0.6296 | 0.5729 |
| 30% | 0.7774 | 0.5875 | 0.5318 |
| 40% | 0.8290 | 0.6630 | 0.6221 |

forget rate, our model achieves the highest average performance across the 15 datasets, suggesting that effectively reducing pseudo-label noise can further enhance detection performance. Therefore, developing more stable pseudo-label refinement strategies is an important direction for future work.

**Ablation Study.** Five distinct model configurations are developed for the ablation experiments, designated as *w/o aux*: Remove auxiliary task during both the model training phase and the testing phase; *w/o contra*: Remove contrastive optimization during the testing phase; *w/o adapt*: Model

updates focus solely on adapting to the features of new data, without considering whether such adaptation may compromise the knowledge previously learned by the model; *w/o TTCL*: Remove entire TTCL module, and TCAD.

As illustrated in Table 4, TCAD achieves state-of-the-art performance. The results of its four variants further demonstrate that: (1) The auxiliary task facilitates the acquisition of richer knowledge, and collaborative dual-task training provides the model with a solid foundation for reliable

Table 4: The average results of the ablation studies across all datasets.

| Metric | *w/o aux* | *w/o contra* | *w/o adapt* | *w/o TTCL* | **TCAD** |
|---|---|---|---|---|---|
| AUC-ROC | 0.6800 | 0.6434 | 0.6590 | 0.7827 | 0.8408 |
| AUC-PR | 0.5548 | 0.5091 | 0.5160 | 0.5487 | 0.6388 |
| F1 | 0.5194 | 0.4609 | 0.4738 | 0.5146 | 0.5953 |

detection capability. (2) Performing contrastive optimization at test time and minimizing the forgetting of previously learned knowledge are both crucial. Detailed results are provided in the Appendix A.7.

**Parameter Sensitivity Analysis.** We conducted a parameter sensitivity analysis with respect to three key factors: the value of K used in the KNN-based contrastive learning module, the confidence threshold for pseudo-label selection, and the weighting coefficients of the adaptation loss and contrastive loss. The resulting performance trends are presented in Figure 6. The main findings from these experiments are summarized as follows: (1) Robustness to the choice of K. The model is robust with respect to the value of K. Under imbalanced normal–abnormal data settings, both AUC-PR and F1 score remain stable. (2) Robustness to the selection rate of high-confidence samples. The model shows stable performance across different selection rates, with the good overall results obtained when the selection rate is set to 10% or 15%. We believe this is because an overly small selection rate provides too few samples to effectively guide test-time adaptation, whereas an overly large selection rate may introduce more incorrectly labeled samples, steering the adaptation process away from the expected direction. (3) Robustness to different loss weight combinations. The model remains stable under different weighting schemes applied to the loss terms.

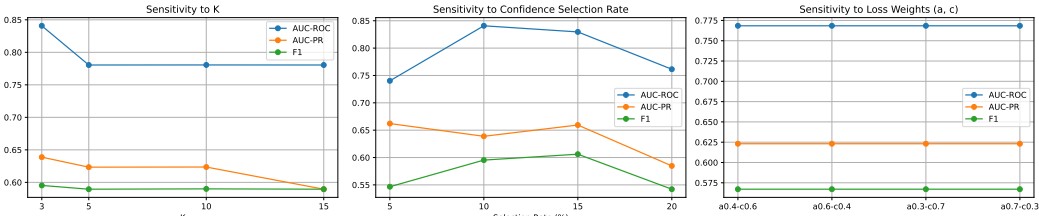

Figure 6: Average detection performance across 15 datasets under different parameter settings

**Computational Cost.** We compare the memory and time overhead of MCM, DRL, and TCAD. The three methods consume similar memory. DRL has the lowest time cost due to efficient representation decomposition, while TCAD takes slightly more time because of model adjustment at the test phase. Detailed results are provided in Appendix A.8.

## 5 CONCLUSION

In this paper, we propose a test-time contrastive learning approach for unsupervised anomaly detection in tabular data, named TCAD. The approach learns rich information from training samples to model normal patterns through Collaborative Dual-task Training. Meanwhile, it employs Test-Time Contrastive Learning to enable the model to adapt to test samples in a designed manner and refine the embedding distribution. Unlike traditional unsupervised anomaly detection methods, TCAD improves the model's ability to identify samples that deviate from learned normal patterns. This is achieved by dynamically updating the model during the test phase using high-confidence samples generated by the trained model. Furthermore, our experiments reveal that: (1) Designing effective model update strategies during the test phase can improve anomaly detection capability. (2) During test-time adaptation in anomaly detection, it is crucial for the model to retain the valuable knowledge acquired during training, while simultaneously avoiding the risk of overfitting to anomalous patterns in the test data. In the future, we aim to explore more effective update strategies during the test phase and investigate the potential of multi-agent approaches for unsupervised anomaly detection.

ETHICS STATEMENT

We affirm that our study has been conducted in full accordance with the ICLR Code of Ethics.

REPRODUCIBILITY STATEMENT

We have taken several steps to ensure the reproducibility of our work. The datasets used in our experiments are all publicly available, and the construction and analysis of data shifts are described in detail in Section 4.1. The implementation details, including model configurations and training hyperparameters, are thoroughly documented in Section 4.2. While the source code is not released during the anonymity period, we plan to make it publicly available after this stage to facilitate future research and ensure proper attribution.

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

# A APPENDIX

## A.1 USE OF LARGE LANGUAGE MODELS (LLMS)

In preparing this manuscript, we employed large language models (LLMs) solely for auxiliary purposes, specifically for checking grammar and improving the clarity of language. Importantly, no LLMs were used in formulating the research motivation, designing the methodology, conducting the experiments, or interpreting the results. All core scientific contributions of this work are entirely original and authored by the researchers.

## A.2 STATISTICS OF DATASETS

The number of samples, feature dimensions, anomaly contamination rates, and category information for the 15 datasets are summarized in Table 5.

Table 5: The statistics of datasets.

| Dataset | Samples | Dim | Anomaly | Category |
|---|---|---|---|---|
| Arrhythmia | 452 | 274 | 66 (15%) | Healthcare |
| BreastW | 683 | 9 | 239 (35%) | Healthcare |
| Cardio | 1831 | 21 | 176 (9.6%) | Healthcare |
| Cardiotocography | 2114 | 21 | 466 (22.04%) | Healthcare |
| Glass | 214 | 9 | 9 (4.2%) | Forensic |
| Ionosphere | 351 | 33 | 126 (36%) | Oryctognosy |
| Mammography | 11183 | 6 | 260 (2.32%) | Healthcare |
| Optdigits | 5216 | 64 | 150 (2.88%) | Image |
| Pendigits | 6870 | 16 | 156 (2.27%) | Image |
| Pima | 768 | 8 | 268 (35%) | Healthcare |
| Satellite | 6435 | 36 | 2036 (32%) | Astronautics |
| Satimage-2 | 5803 | 36 | 71 (1.2%) | Astronautics |
| Thyroid | 3772 | 6 | 93 (2.5%) | Healthcare |
| Wbc | 278 | 30 | 21 (5.6%) | Healthcare |
| Wine | 129 | 13 | 10 (7.75%) | Chemistry |

## A.3 DATA SHIFT ANALYSIS

The visualizations of each dataset's overall distribution and the distribution of the i-th feature are presented in Figure 7. The results of Jeffreys Divergence (JD) and the Optimal Transport Dataset Distance (OTDD) for each dataset are summarized in Table 6.

Table 6: The results of Jeffreys Divergence (JD) and Optimal Transport Dataset Distance (OTDD) between the distributions of normal samples in the training and test sets across all datasets.

| Metrics / Dataset | arrhythmia | breastw | cardio | cardiotocography | glass |
|---|---|---|---|---|---|
| JD | 1.39 | 0.48 | 1.72 | 0.46 | 1.28 |
| OTDD | 0.31 | 0.24 | 0.29 | 0.14 | 0.15 |
| **Metrics / Dataset** | **ionosphere** | **mammography** | **optdigits** | **pendigits** | **pima** |
| JD | 9.08 | 2.40 | 0.19 | 0.80 | 1.65 |
| OTDD | 0.39 | 0.09 | 0.55 | 0.49 | 0.16 |
| **Metrics / Dataset** | **satellite** | **satimage-2** | **thyroid** | **wbc** | **wine** |
| JD | 3.23 | 2.83 | 0.37 | 4.17 | 15.43 |
| OTDD | 0.01 | 0.33 | 0.24 | 0.25 | 0.27 |

## A.4 COMPUTATION OF CONFIDENCE INTERVALS FOR THE MAIN METRICS

To provide a robust estimate of the variability in our results across datasets, we compute bootstrap confidence intervals for the main evaluation metrics. Specifically, for each metric, we perform 1,000 bootstrap resamples over the dataset-level scores. In each iteration, we randomly sample with replacement from the scores and compute the mean of the resampled set. The 95% confidence interval is then obtained by taking the 2.5th and 97.5th percentiles of the bootstrapped mean values. This procedure ensures that the reported mean performance is accompanied by a statistically meaningful

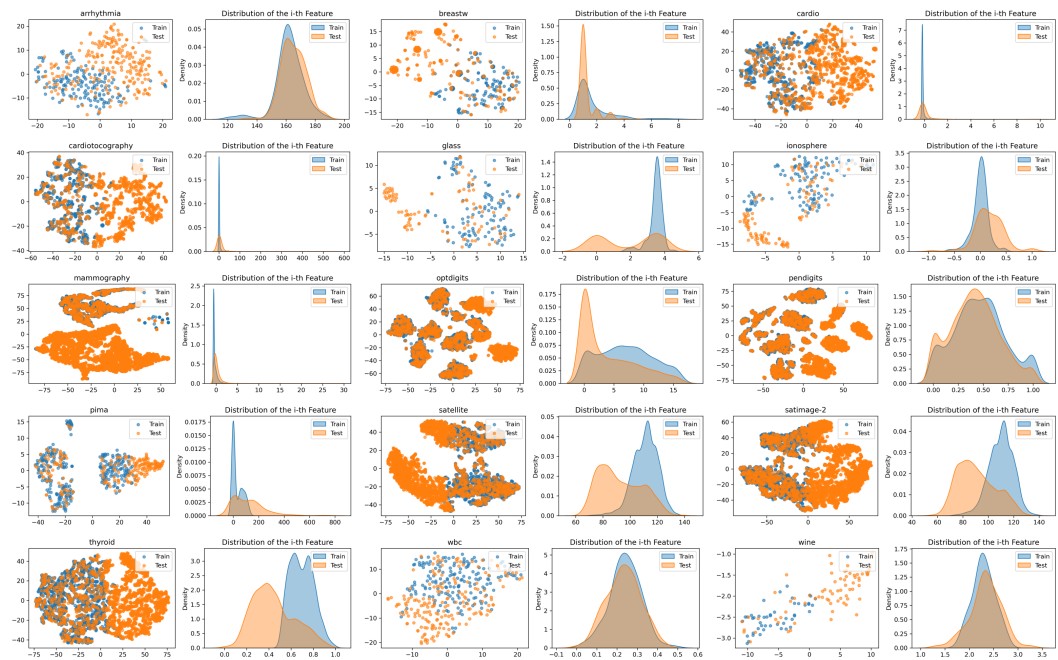

Figure 7: The comparison of normal sample distributions between the training and test sets for each dataset, along with the distribution comparison of the $i - th$ feature.

measure of uncertainty, reflecting the variability across datasets. The detailed results are shown in Table 7.

Table 7: Confidence Intervals of Main Metrics for Each Method.

|        | Iforest            | LOF                | OCSVM              | DeepSVDD           | ECOD               |
| ------ | ------------------ | ------------------ | ------------------ | ------------------ | ------------------ |
| auc-roc | [0.5847, 0.7246] | [0.5937, 0.6754] | [0.5433, 0.6641] | [0.5490, 0.6554] | [0.6055, 0.7252] |
| auc-pr  | [0.4561, 0.6489] | [0.5237, 0.6524] | [0.5289, 0.6632] | [0.4153, 0.6242] | [0.4152, 0.6204] |
| f1      | [0.2822, 0.5360] | [0.2525, 0.4877] | [0.2578, 0.4967] | [0.2216, 0.4703] | [0.3267, 0.5329] |
|        | GOAD               | NeuTraLAD          | ICL                | DIF                | SLAD               |
| auc-roc | [0.6342, 0.8030] | [0.6010, 0.7583] | [0.6351, 0.7982] | [0.7096, 0.8774] | [0.6661, 0.8171] |
| auc-pr  | [0.3420, 0.6320] | [0.2711, 0.5722] | [0.3415, 0.6304] | [0.4317, 0.6971] | [0.3352, 0.6624] |
| f1      | [0.3093, 0.5948] | [0.2425, 0.5514] | [0.3212, 0.6025] | [0.4003, 0.6499] | [0.2997, 0.6133] |
|        | LUNAR              | MCM                | DRL                | TCAD(ours)         |                    |
| auc-roc | [0.6038, 0.7408] | [0.7277, 0.8664] | [0.6255, 0.7874] | [0.7685, 0.9111] |                  |
| auc-pr  | [0.5487, 0.6790] | [0.4138, 0.7196] | [0.2834, 0.6004] | [0.5050, 0.7717] |                  |
| f1      | [0.2585, 0.5164] | [0.3290, 0.6384] | [0.2465, 0.5570] | [0.4708, 0.7149] |                  |

## A.5 COMPETING METHODS

The detailed introduction of each method is as follows:

- **IForest** (Liu et al., 2008) isolates anomalies by recursively partitioning the data using random splits. The core idea is that anomalies are easier to isolate due to their distinctiveness, requiring fewer partitions compared to normal data points, and this isolation process is used to identify anomalies.

- **LOF** (Breunig et al., 2000) evaluates the local density of data points by comparing the density of a point with that of its neighbors. Points with significantly lower density than their neighbors are considered anomalies, as they deviate from the expected local structure of the data.

- **OCSVM** (Schölkopf et al., 1999) constructs a hyperplane in a high-dimensional space that maximizes the margin around the normal data. This results in the majority of data points being mapped within the boundary, while points that deviate significantly from this boundary are identified as anomalies.

- **DeepSVDD** (Ruff et al., 2018) learns a deep feature representation of the data while simultaneously minimizing the volume of a hypersphere that encloses the normal data. Data points that lie outside this learned hypersphere are detected as anomalies.

- **ECOD** (Li et al., 2022) leverages the empirical cumulative distribution function (ECDF) to detect anomalies. For each feature in the dataset, ECOD computes the ECDF, which captures the data's distributional properties in a robust and interpretable manner. Points that fall in the extreme tails of the distribution are assigned higher anomaly scores.

- **GOAD** (Bergman & Hoshen, 2020) generalizes the class of transformation functions to include affine transformation which allows it to generalize to non-image data. By applying these transformations to the input data, GOAD trains a classifier to distinguish between the transformed versions. At test time, normal data will exhibit predictable patterns under these transformations, while abnormal data fails to conform to these patterns, making it easier to be identified.

- **NeuTral AD** (Qiu et al., 2021) learns a set of neural transformations, parameterized by neural networks, which map the input data to various transformed spaces and capture the intrinsic structure of normal data. During the testing phase, samples that do not follow the learned patterns are detected as anomalies.

- **ICL** (Shenkar & Wolf, 2022) employs contrastive loss to learn mappings that maximize the mutual information between each sample and the part that is masked out and capture the structure of the samples of the single training class. Test samples are scored by measuring whether the learned mappings lead to a small contrastive loss using the masked parts of this sample. Samples with high loss values are regarded as anomalies.

- **DIF** (Xu et al., 2023a) uses randomly initialized neural networks to create random representation ensembles. Through random axis-parallel cuts on these representations, it realizes nonlinear partitioning in the original space. With CERE for efficient feature mapping and DEAS combining path length and feature deviation, DIF scores anomalies via isolation tree ensembles.

- **SLAD** (Xu et al., 2023b) introduces scale learning for tabular anomaly detection, defining "scale" as the dimensionality relationship between data sub-vectors and their representations. It uses a neural network to learn distribution alignment of subspace transformations via Jensen-Shannon divergence loss, modeling inlier structural regularities. Test instances are scored by divergence from learned scale distributions, high loss indicates anomalies.

- **LUNAR** (Goodge et al., 2022) reframes local outlier detection as a GNN message-passing problem, where samples are nodes connected to k-nearest neighbors. It replaces fixed aggregation rules with learnable neural aggregation and trains with synthetic negatives, enabling adaptive, robust anomaly detection.

- **MCM** (Yin et al., 2024) adapts mask modeling to address the problem of tabular data anomaly detection. Mask generator and autoencoder are employed to capture intrinsic correlations between features existing in training tabular data and model the "characteristic patterns" by such correlations. Samples that deviate from these correlations are predicted as anomalies.

- **DRL** (Ye et al., 2025) tackles tabular anomaly detection by mapping data into a constrained latent space, where each normal sample is represented as a weighted linear combination of fixed orthogonal basis vectors. It enhances discriminability by increasing the variance of normal weights and preserves feature correlations via alignment loss.

## A.6    FULL COMPARISON RESULTS WITH BASELINE METHODS

The detailed results of the main experiments are presented in Table 8, 9 and 10.

Table 8: Comparison of AUC-PR(↑)results between baseline methods and TCAD on 15 datasets.

| | Iforest | LOF | OCSVM | DeepSVDD | ECOD | GOAD | NeuTraLAD | ICL | DIF | SLAD | LUNAR | MCM | DRL | TCAD(ours) |
|---|---|---|---|---|---|---|---|---|---|---|---|---|---|---|
| arrhythmia | 0.6019 | 0.5676 | 0.6111 | 0.6115 | 0.6244 | 0.5867 | 0.5023 | 0.5407 | 0.6294 | 0.5372 | 0.5856 | 0.5657 | 0.5510 | 0.6212 |
| breastw | 0.8536 | 0.7818 | 0.7656 | 0.8256 | 0.7581 | 0.9860 | 0.5662 | 0.8508 | 0.9737 | 0.9569 | 0.9644 | 0.9911 | 0.9302 | 0.9921 |
| cardio | 0.5381 | 0.5581 | 0.5835 | 0.4407 | 0.6860 | 0.4606 | 0.2458 | 0.3687 | 0.6176 | 0.4277 | 0.4782 | 0.6849 | 0.4054 | 0.6385 |
| cardiotocography | 0.5529 | 0.5562 | 0.6531 | 0.5417 | 0.6120 | 0.3408 | 0.3746 | 0.4113 | 0.4944 | 0.3255 | 0.4604 | 0.4051 | 0.3682 | 0.3906 |
| glass | 0.1721 | 0.3482 | 0.3472 | 0.5118 | 0.1658 | 0.0994 | 0.1201 | 0.2309 | 0.1013 | 0.1105 | 0.5750 | 0.1099 | 0.1191 | 0.1504 |
| ionosphere | 0.8094 | 0.8032 | 0.8094 | 0.7772 | 0.6842 | 0.6543 | 0.6337 | 0.6063 | 0.8097 | 0.7038 | 0.8241 | 0.7504 | 0.8282 | 0.7552 |
| mammography | 0.2713 | 0.4927 | 0.4582 | 0.4812 | 0.4862 | 0.3293 | 0.0563 | 0.1560 | 0.4507 | 0.1467 | 0.4962 | 0.4781 | 0.1022 | 0.5407 |
| optdigits | 0.3311 | 0.5605 | 0.5290 | 0.0348 | 0.0373 | 0.0640 | 0.0528 | 0.0725 | 0.0647 | 0.0690 | 0.5526 | 0.1103 | 0.0997 | 0.4199 |
| pendigits | 0.3430 | 0.5283 | 0.5162 | 0.1523 | 0.4147 | 0.0445 | 0.0362 | 0.0431 | 0.5583 | 0.0475 | 0.5288 | 0.0430 | 0.0333 | 0.0964 |
| pima | 0.7448 | 0.7441 | 0.7763 | 0.6826 | 0.7113 | 0.5587 | 0.5373 | 0.5787 | 0.5943 | 0.5902 | 0.6611 | 0.5707 | 0.5347 | 0.5921 |
| satellite | 0.7169 | 0.7177 | 0.7173 | 0.6894 | 0.6437 | 0.7595 | 0.8306 | 0.7927 | 0.7173 | 0.8339 | 0.6926 | 0.8199 | 0.8400 | 0.8307 |
| satimage-2 | 0.5006 | 0.5013 | 0.5094 | 0.5129 | 0.5931 | 0.6819 | 0.8071 | 0.9461 | 0.9754 | 0.9588 | 0.5114 | 0.9717 | 0.0925 | 0.9716 |
| thyroid | 0.6202 | 0.4433 | 0.4208 | 0.3091 | 0.5818 | 0.1503 | 0.1919 | 0.1677 | 0.2379 | 0.4751 | 0.5032 | 0.2925 | 0.0629 | 0.7773 |
| wbc | 0.6235 | 0.5745 | 0.6175 | 0.5703 | 0.5609 | 0.5047 | 0.2311 | 0.5735 | 0.4594 | 0.2623 | 0.6094 | 0.7268 | 0.4460 | 0.8147 |
| wine | 0.5627 | 0.5781 | 0.5610 | 0.5641 | 0.3177 | 0.9909 | 0.9667 | 0.8813 | 0.8112 | 0.9573 | 0.6667 | 0.9909 | 0.9430 | 0.9909 |
| Average PR | 0.5494 | 0.5837 | 0.5917 | 0.5136 | 0.5251 | 0.4807 | 0.4101 | 0.4813 | 0.5663 | 0.4934 | 0.6073 | 0.5674 | 0.4237 | 0.6388 |
| Average Ranking | 6.6 | 6.0 | 6.0 | 7.86 | 7.33 | 9.26 | 11.06 | 9.2 | 6.4 | 8.86 | 5.53 | 6.6 | 9.86 | 4.26 |

Table 9: Comparison of AUC-ROC(↑)results between baseline methods and TCAD on 15 datasets.

| | Iforest | LOF | OCSVM | DeepSVDD | ECOD | GOAD | NeuTraLAD | ICL | DIF | SLAD | LUNAR | MCM | DRL | TCAD(ours) |
|---|---|---|---|---|---|---|---|---|---|---|---|---|---|---|
| arrhythmia | 0.7229 | 0.6797 | 0.5 | 0.5022 | 0.7175 | 0.7694 | 0.7127 | 0.7115 | 0.8167 | 0.7227 | 0.7089 | 0.7629 | 0.7345 | 0.8190 |
| breastw | 0.8111 | 0.6469 | 0.5973 | 0.7557 | 0.5725 | 0.9814 | 0.6669 | 0.8860 | 0.9640 | 0.9467 | 0.9609 | 0.9936 | 0.9660 | 0.9933 |
| cardio | 0.5943 | 0.6497 | 0.6929 | 0.6019 | 0.8388 | 0.6489 | 0.5901 | 0.6479 | 0.9244 | 0.77109 | 0.5633 | 0.6849 | 0.6418 | 0.8199 |
| cardiotocography | 0.4837 | 0.4760 | 0.5018 | 0.5480 | 0.6695 | 0.4619 | 0.4829 | 0.4679 | 0.7682 | 0.4554 | 0.4094 | 0.6476 | 0.4843 | 0.6402 |
| glass | 0.4249 | 0.5437 | 0.5384 | 0.6466 | 0.5236 | 0.5390 | 0.6194 | 0.6147 | 0.4031 | 0.6017 | 0.7287 | 0.6190 | 0.6076 | 0.7021 |
| ionosphere | 0.6587 | 0.6407 | 0.6587 | 0.5501 | 0.5664 | 0.6288 | 0.6342 | 0.6371 | 0.7559 | 0.6885 | 0.7048 | 0.7885 | 0.8112 | 0.7446 |
| mammography | 0.4956 | 0.5918 | 0.8010 | 0.5835 | 0.7321 | 0.8618 | 0.6944 | 0.8014 | 0.8561 | 0.7547 | 0.5898 | 0.8635 | 0.7139 | 0.8970 |
| optdigits | 0.6219 | 0.7765 | 0.5 | 0.465 | 0.4839 | 0.5949 | 0.4924 | 0.645 | 0.5466 | 0.6240 | 0.7386 | 0.8295 | 0.7441 | 0.9460 |
| pendigits | 0.6461 | 0.6945 | 0.8033 | 0.5252 | 0.6298 | 0.6016 | 0.5021 | 0.5763 | 0.9446 | 0.5895 | 0.6666 | 0.6447 | 0.4586 | 0.7313 |
| pima | 0.5626 | 0.5662 | 0.5 | 0.4893 | 0.4893 | 0.5769 | 0.5122 | 0.4813 | 0.5084 | 0.5574 | 0.5579 | 0.5251 | 0.6127 | 0.4798 |
| satellite | 0.5634 | 0.5694 | 0.5 | 0.6331 | 0.6086 | 0.7155 | 0.7912 | 0.7756 | 0.6738 | 0.8006 | 0.5747 | 0.8065 | 0.7992 | 0.8143 |
| satimage-2 | 0.6724 | 0.6866 | 0.5 | 0.8799 | 0.9124 | 0.9640 | 0.9960 | 0.9958 | 0.9973 | 0.9972 | 0.5893 | 0.9986 | 0.7863 | 0.9985 |
| thyroid | 0.9196 | 0.5951 | 0.6239 | 0.5618 | 0.8092 | 0.6534 | 0.7277 | 0.6166 | 0.8838 | 0.8444 | 0.6500 | 0.7696 | 0.5262 | 0.9630 |
| wbc | 0.8289 | 0.7866 | 0.8372 | 0.7382 | 0.7694 | 0.9032 | 0.7881 | 0.9234 | 0.8877 | 0.8139 | 0.7995 | 0.9643 | 0.8887 | 0.9723 |
| wine | 0.7736 | 0.6250 | 0.5 | 0.5278 | 0.5875 | 0.9986 | 0.9931 | 0.9889 | 0.9542 | 0.9944 | 0.8611 | 0.9988 | 0.9931 | 0.9986 |
| Average ROC | 0.6519 | 0.6352 | 0.6036 | 0.6005 | 0.6665 | 0.7223 | 0.6781 | 0.7197 | 0.7955 | 0.7401 | 0.6713 | 0.7989 | 0.7090 | 0.8408 |
| Average Ranking | 8.86 | 9.26 | 9.66 | 11.0 | 8.66 | 7.06 | 9.2 | 7.93 | 5.06 | 6.6 | 8.86 | 3.0 | 7.53 | 2.06 |

Table 10: Comparison of F1(↑)results between baseline methods and TCAD on 15 datasets.

| | Iforest | LOF | OCSVM | DeepSVDD | ECOD | GOAD | NeuTraLAD | ICL | DIF | SLAD | LUNAR | MCM | DRL | TCAD(ours) |
|---|---|---|---|---|---|---|---|---|---|---|---|---|---|---|
| arrhythmia | 0.5466 | 0.4842 | 0.3636 | 0.3646 | 0.5691 | 0.5909 | 0.5303 | 0.4697 | 0.5909 | 0.5152 | 0.5325 | 0.5 | 0.5 | 0.5455 |
| breastw | 0.8284 | 0.721 | 0.6938 | 0.7888 | 0.6809 | 0.9665 | 0.6360 | 0.7876 | 0.9436 | 0.8852 | 0.9590 | 0.9540 | 0.9205 | 0.9582 |
| cardio | 0.2964 | 0.3319 | 0.3628 | 0.3051 | 0.6509 | 0.5170 | 0.2670 | 0.4545 | 0.5909 | 0.4830 | 0.2773 | 0.3239 | 0.3864 | 0.6023 |
| cardiotocography | 0.4218 | 0.4203 | 0.4704 | 0.4433 | 0.5435 | 0.3004 | 0.3305 | 0.3004 | 0.5365 | 0.2961 | 0.3374 | 0.3584 | 0.3348 | 0.4099 |
| glass | 0.0870 | 0.1724 | 0.1695 | 0.2192 | 0.1250 | 0 | 0 | 0.2222 | 0.1111 | 0 | 0.2609 | 0 | 0 | 0.1111 |
| ionosphere | 0.7654 | 0.7561 | 0.7654 | 0.7143 | 0.5398 | 0.5635 | 0.6190 | 0.5952 | 0.6746 | 0.6270 | 0.7871 | 0.6349 | 0.7143 | 0.6508 |
| mammography | 0.0664 | 0.0853 | 0.2836 | 0.0839 | 0.1340 | 0.4154 | 0.0038 | 0.2192 | 0.4692 | 0.1577 | 0.0848 | 0.4923 | 0.1 | 0.5407 |
| optdigits | 0.1709 | 0.2158 | 0.1095 | 0.0059 | 0.0081 | 0 | 0 | 0.02 | 0.04 | 0 | 0.1905 | 0.02 | 0.0133 | 0.4733 |
| pendigits | 0.1317 | 0.1181 | 0.1906 | 0.0817 | 0.3574 | 0 | 0 | 0.0256 | 0.5256 | 0.0385 | 0.1089 | 0.0064 | 0.0192 | 0.1154 |
| pima | 0.6689 | 0.6667 | 0.7118 | 0.5709 | 0.5738 | 0.5746 | 0.5522 | 0.5448 | 0.5821 | 0.5784 | 0.4847 | 0.5149 | 0.5373 | 0.5858 |
| satellite | 0.6261 | 0.6286 | 0.6059 | 0.6144 | 0.5171 | 0.6051 | 0.7194 | 0.6685 | 0.5953 | 0.7083 | 0.6102 | 0.7141 | 0.7269 | 0.7210 |
| satimage-2 | 0.0561 | 0.0585 | 0.0369 | 0.1723 | 0.4710 | 0.6620 | 0.8592 | 0.9014 | 0.9577 | 0.9014 | 0.0446 | 0.9296 | 0.1268 | 0.9296 |
| thyroid | 0.4674 | 0.0975 | 0.1084 | 0.0942 | 0.5660 | 0.1398 | 0.1828 | 0.1613 | 0.1935 | 0.4516 | 0.1093 | 0.2903 | 0.0430 | 0.6667 |
| wbc | 0.3962 | 0.3725 | 0.4301 | 0.3077 | 0.5306 | 0.5238 | 0.2381 | 0.6667 | 0.5238 | 0.1905 | 0.3590 | 0.6667 | 0.5238 | 0.7143 |
| wine | 0.5 | 0.2703 | 0.2174 | 0.2273 | 0.2727 | 0.9 | 0.9 | 0.9 | 0.6 | 0.9 | 0.5 | 0.9 | 0.9 | 0.9 |
| Average F1 | 0.4019 | 0.3599 | 0.3679 | 0.3329 | 0.4359 | 0.4506 | 0.3892 | 0.4624 | 0.5289 | 0.4488 | 0.3764 | 0.4870 | 0.3897 | 0.5953 |
| Average Ranking | 6.93 | 7.8 | 7.73 | 9.4 | 7.0 | 7.86 | 10.13 | 7.46 | 4.93 | 8.13 | 8.0 | 6.86 | 8.33 | 3.6 |

## A.7 RESULTS OF ABLATION STUDY

The detailed results of the ablation study are presented in Table 11.

Table 11: The evaluation results of ablation experiments across the datasets.

| Dataset | *w/o aux* | | | *w/o contra* | | | *w/o adapt* | | | *w/o TTCL* | | | TCAD | | |
|---|---|---|---|---|---|---|---|---|---|---|---|---|---|---|---|
| | auc-roc | auc-pr | f1 | auc-roc | auc-pr | f1 | auc-roc | auc-pr | f1 | auc-roc | auc-pr | f1 | auc-roc | auc-pr | f1 |
| arrhythmia | 0.5808 | 0.5295 | 0.4394 | 0.6195 | 0.5795 | 0.5152 | 0.4509 | 0.4032 | 0.303 | 0.7437 | 0.5612 | 0.5 | 0.819 | 0.6212 | 0.5455 |
| breastw | 0.9833 | 0.9882 | 0.9414 | 0.9856 | 0.9897 | 0.9582 | 0.9841 | 0.9886 | 0.9498 | 0.99 | 0.9882 | 0.9498 | 0.9933 | 0.9921 | 0.9582 |
| cardio | 0.7744 | 0.6366 | 0.5966 | 0.7283 | 0.6246 | 0.5966 | 0.7309 | 0.6382 | 0.5852 | 0.7106 | 0.2561 | 0.2273 | 0.8199 | 0.6385 | 0.6023 |
| cardiotocography | 0.3707 | 0.3575 | 0.279 | 0.3122 | 0.3368 | 0.2725 | 0.2981 | 0.3231 | 0.2618 | 0.5941 | 0.3853 | 0.3691 | 0.6402 | 0.3906 | 0.4099 |
| glass | 0.2726 | 0.0858 | 0.1111 | 0.083 | 0.0649 | 0 | 0.2756 | 0.0797 | 0 | 0.6974 | 0.1596 | 0.2222 | 0.7021 | 0.1504 | 0.1111 |
| ionosphere | 0.5012 | 0.5321 | 0.627 | 0.7009 | 0.7124 | 0.6905 | 0.4966 | 0.53 | 0.6429 | 0.715 | 0.7432 | 0.6349 | 0.7446 | 0.7552 | 0.6508 |
| mammography | 0.7249 | 0.1763 | 0.2577 | 0.7954 | 0.1955 | 0.2769 | 0.8856 | 0.5099 | 0.5692 | 0.8658 | 0.5074 | 0.5192 | 0.897 | 0.5407 | 0.5462 |
| optdigits | 0.2723 | 0.037 | 0 | 0.3683 | 0.0425 | 0 | 0.6349 | 0.0705 | 0 | 0.8171 | 0.1337 | 0.0733 | 0.946 | 0.4199 | 0.4733 |
| pendigits | 0.9471 | 0.7146 | 0.6731 | 0.7514 | 0.0814 | 0 | 0.8837 | 0.2974 | 0.3526 | 0.4142 | 0.0378 | 0.0192 | 0.7313 | 0.0964 | 0.1154 |
| pima | 0.4721 | 0.5484 | 0.4851 | 0.478 | 0.55 | 0.4813 | 0.4704 | 0.5493 | 0.4851 | 0.592 | 0.6311 | 0.6045 | 0.5724 | 0.5921 | 0.5858 |
| satellite | 0.6171 | 0.7131 | 0.5319 | 0.4813 | 0.7486 | 0.5648 | 0.5581 | 0.6679 | 0.4641 | 0.8054 | 0.8409 | 0.7279 | 0.8143 | 0.8307 | 0.721 |
| satimage-2 | 0.9876 | 0.945 | 0.9014 | 0.9855 | 0.9383 | 0.9014 | 0.9934 | 0.9385 | 0.8794 | 0.9983 | 0.9718 | 0.9296 | 0.9985 | 0.9716 | 0.9296 |
| thyroid | 0.7541 | 0.3503 | 0.3333 | 0.4189 | 0.0638 | 0.043 | 0.2801 | 0.0349 | 0 | 0.864 | 0.3749 | 0.3763 | 0.963 | 0.7773 | 0.6667 |
| wbc | 0.9448 | 0.718 | 0.7143 | 0.9448 | 0.718 | 0.7143 | 0.9448 | 0.718 | 0.7143 | 0.9356 | 0.6493 | 0.6667 | 0.9723 | 0.8147 | 0.7143 |
| wine | 0.9983 | 0.9909 | 0.9 | 0.9983 | 0.9909 | 0.9 | 0.9983 | 0.9909 | 0.9 | 0.9986 | 0.9909 | 0.9 | 0.9986 | 0.9909 | 0.9 |
| Average value | 0.6800 | 0.5548 | 0.5194 | 0.6434 | 0.5091 | 0.4609 | 0.6590 | 0.5160 | 0.4738 | 0.7827 | 0.5487 | 0.5146 | 0.8408 | 0.6388 | 0.5953 |

## A.8 COMPARISON RESULTS OF COMPUTATIONAL COST

The detailed results of computational cost comparison are presented in Table 12.

Table 12: The evaluation results of computational cost across the datasets.

| Dataset | Memory Usage | | | Time Overhead | | |
|---|---|---|---|---|---|---|
| | MCM | DRL | TCAD | MCM | DRL | TCAD |
| arrhythmia | 1707 | 2042 | 2053 | 15 | 6 | 26 |
| breastw | 1563 | 2046 | 2237 | 8 | 4 | 41 |
| cardio | 1739 | 2046 | 2211 | 18 | 5 | 44 |
| cardiotocography | 1705 | 2047 | 2102 | 16 | 8 | 59 |
| glass | 1564 | 2047 | 1949 | 9 | 3 | 21 |
| ionosphere | 1623 | 2047 | 2289 | 11 | 3 | 50 |
| mammography | 1961 | 1786 | 2426 | 81 | 22 | 121 |
| optdigits | 1911 | 1627 | 2460 | 48 | 11 | 320 |
| pendigits | 1927 | 1711 | 2762 | 54 | 15 | 360 |
| pima | 1562 | 2039 | 2336 | 10 | 5 | 60 |
| satellite | 1912 | 1628 | 2657 | 48 | 11 | 290 |
| satimage-2 | 1932 | 1672 | 2150 | 58 | 13 | 47 |
| thyroid | 1888 | 1564 | 2510 | 40 | 9 | 160 |
| wbc | 1563 | 2039 | 2069 | 10 | 5 | 26 |
| wine | 1562 | 2040 | 2071 | 10 | 5 | 28 |
| Average value | 1741 | 1892 | 2285 | 29 | 8.3 | 110.2 |

