# OpenReview forum: "Unsupervised Anomaly Detection in Tabular Data with Test-time Contrastive Learning"
_ICLR.cc/2026/Conference — Submitted to ICLR 2026_

### Official Review · Reviewer_jpNR · 2025-10-25

**Soundness:** 2
**Presentation:** 2
**Contribution:** 2
**Rating:** 4
**Confidence:** 4

**Summary:**

The paper presents TCAD, a test-time contrastive learning approach for unsupervised anomaly detection in tabular data, where normal patterns at test time deviate from those observed during training. The method has two stages. First, a collaborative dual-task training stage learns low-level and high-level features using a masked autoencoder as the primary task and an embedding reconstruction as the auxiliary task. Second, at test time, the method assigns pseudo labels to high-confidence samples, adapts to pseudo-normal examples while discouraging accurate modeling of pseudo-anomalous ones, and runs a K-nearest neighbors contrastive objective that pulls pseudo-normal embeddings toward the training distribution and pushes pseudo-anomalous ones away. The overall loop iterates until all test samples receive pseudo labels. A clear motivation and a helpful overview are provided. The authors construct distribution shifts on fifteen ODDS and ADBench datasets by clustering normals using K-means, training on the largest cluster, and testing on the remainder, which is mixed with anomalies.  The headline results show average improvements over the strongest baseline. The ablations indicate that both the auxiliary task and the test time contrastive component matter, while the cost table shows that TCAD has a noticeably higher time overhead than DRL and MCM on average.

I appreciate the practical problem and the clean formulation.  At the same time, I have concerns about several choices that affect the strength of the claims. The shift construction with K-means on normals may bias training toward a single mode and therefore create an easier adaptation target than many real deployments where shifts arise from covariate drift, concept drift, or temporal regimes. Reliance on a known contamination rate is a strong assumption; in practice, this value is rarely known and often misspecified, yet the method uses it for high-confidence sample selection and for final thresholding. The test time loop selects extreme samples based on the model itself, which can amplify confirmation bias when early pseudo labels are wrong. The K nearest neighbors contrastive step uses a fixed k  and a fixed temperature and does not study sensitivity to these choices.

**Strengths:**

The paper focuses on a realistic setting where normal behavior at test time shifts relative to training. The design is simple and implementable in standard toolchains, and the figures and equations are accessible. The empirical sweep across fifteen datasets is helpful, and the per-dataset tables make it easy to identify where the gains originate. The ablation study supports the role of the auxiliary task and the contrastive step, and the significance tests for F1 add credibility. I also found the visual and quantitative shift analysis useful for readers who may want to reproduce the construction on other corpora.

**Weaknesses:**

The shift protocol may not reflect many real-world patterns. Training on the largest normal cluster and testing on the remainder can privilege cluster structure and does not cover temporal drift or label shift scenarios. The method assumes a known contamination rate during selection and final thresholding; this is a strong requirement, and the paper does not evaluate robustness when this value is wrong. The test time loop depends on the model to select extremes, which can create a feedback effect. The small pseudo-label audit in Table 1 is encouraging for four datasets, but a broader and more systematic analysis is lacking. Several hyperparameters are fixed globally, for example, k equal to three in the K nearest neighbors step, and there is no sensitivity study for these or for the balance between the adaptation and the contrastive losses. Baseline tuning appears uneven across families, and the search spaces are not aligned, which can inflate the advantage. The summary plots do not show confidence intervals for AUC metrics, and many experiments are averaged over only three seeds.

**Questions:**

1) Could you report robustness to misspecified contamination rates and provide a variant that estimates or adapts this value from data rather than assuming it.

2) How sensitive are results to k in the contrastive step, to the selection batch size per round, and to the trade-off between adaptation and contrastive losses?

3) Would you consider a stronger shift protocol, for example, temporally stratified splits or the AnoShift design with time evolving normals, and report the same tables?

4) Can you add confidence intervals for all main metrics and broaden the pseudo-label noise analysis beyond the four datasets with a simple noise control, like co-teaching or small disagreement filtering?

5) Can you present an appropriate cost comparison that includes end-to-end latency per test sample through all adaptation rounds, and discuss memory growth of the neighbor pool as it accumulates known normals?

---

> ### Author Response · Authors · 2025-12-02
> **Responses to Questions #1, #2 and #3**
>
> > **Question #1** Could you report robustness to misspecified contamination rates and provide a variant that estimates or adapts this value from data rather than assuming it.
>
> Yes, we could. We have added experiments with different contamination rates (5%, 10%, 15%, and 20%) and also provided a variant that adaptively estimates the contamination rate.
> Additionally, we would like to clarify that the prior contamination rate is not used during model training or test-time adaptation, but is only employed for computing the F1 score during performance evaluation.\
> For a detailed response, please refer to our responses to Reviewer yiGT’s Weakness #2 and Question #2, which contain a comprehensive description of the experimental setup, results, and analysis.
>
> > **Question #2** How sensitive are results to k in the contrastive step, to the selection batch size per round, and to the trade-off between adaptation and contrastive losses?
>
> Thank you for your question. We have conducted a sensitivity analysis on the relevant parameters, and the results are presented in Tables 1–3. The main findings from these experiments are summarized as follows:
> 1. **Robustness to the choice of K.**
> The model is robust with respect to the value of K. Under imbalanced normal–abnormal data settings, both AUC-PR and F1 score remain stable.
> 2. **Robustness to the selection rate of high-confidence samples.**
> The model shows stable performance across different selection rates, with good overall results obtained when the selection rate is set to 10% or 15%. We believe this is because an overly small selection rate provides too few samples to effectively guide test-time adaptation, whereas an overly large selection rate may introduce more incorrectly labeled samples, steering the adaptation process away from the expected direction.
> 3. **Robustness to different loss weight combinations.**
> The model remains stable under different weighting schemes applied to the loss terms.
> ```
> Table 1. Average Detection Performance Across 15 Datasets Under Different Values of K.
> ```
> | K/Metric | auc-roc | auc-pr | f1 |
> | --- | --- | --- | --- |
> | 3 | 0.8408 | 0.6388 | 0.5953 |
> | 5 | 0.7805 | 0.6234 | 0.5895 |
> | 10 | 0.7806 | 0.6236 | 0.5900 |
> | 15 | 0.7805 | 0.6234 | 0.5895 |
>
> ```
> Table 2. Average Detection Performance Across 15 Datasets Under Different Confidence Selection Rates.
> ```
> | Selection rate/Metric | auc-roc | auc-pr | f1 |
> | --- | --- | --- | --- |
> | 5% | 0.7402 | 0.6621 | 0.5466 |
> | 10% | 0.8408 | 0.6388 | 0.5953 |
> | 15% | 0.8296 | 0.6594 | 0.6061 |
> | 20% | 0.7614 | 0.5847 | 0.5421 |
>
> ```
> Table 3. Average Detection Performance Across 15 Datasets Under Different Loss Weight Combinations (a: weight of adaptation loss, c: weight of contrastive loss).
> ```
> | Weight/Metric | auc-roc | auc-pr | f1 |
> | --- | --- | --- | --- |
> | a0.4-c0.6 | 0.7684 | 0.6231 | 0.5671 |
> | a0.6-c0.4 | 0.7684 | 0.6231 | 0.5671 |
> | a0.3-c0.7 | 0.7684 | 0.6231 | 0.5671 |
> | a0.7-c0.3 | 0.7684 | 0.6231 | 0.5671 |
>
> > **Question #3** Would you consider a stronger shift protocol, for example, temporally stratified splits or the AnoShift design with time evolving normals, and report the same tables?
>
> Yes, we would. We have conducted experiments using subsets of the Anoshift benchmark. Specifically, due to the limitations of computational resources, we use the 2006–2010 valid sets as the training data and the 2011–2015 valid sets as the test data.
> Since each valid dataset in Anoshift follows the same distribution as its corresponding full-year dataset, this setup effectively simulates the distribution shift caused by temporal evolution using data at a smaller scale.
>
> Following the evaluation protocol of Anoshift, we report AUC-ROC values. As shown in Table 4, MCM only outperforms TCAD on the 2011 test set, which is closest to the training years. As the distribution shift becomes more pronounced over time, TCAD consistently surpasses MCM on the 2012–2015 test sets. The largest improvement is observed in 2014, where TCAD exceeds MCM by 0.1115. These results demonstrate that TCAD maintains strong and competitive detection performance under distribution shifts induced by temporal evolution.
> ```
> Table 4. AUC-ROC Comparison Between MCM and TCAD on the Anoshift Subsets.
> ```
> | Method/Year | 2011 | 2012 | 2013 | 2014 | 2015 |
> | --- | --- | --- | --- | --- | --- |
> | MCM | 0.9445 | 0.8381 | 0.8341 | 0.3620 | 0.2963 |
> | TCAD | 0.8962	| 0.9038 | 0.8692 | 0.4735 | 0.3778 |

---

> ### Author Response · Authors · 2025-12-02
> **Responses to Questions #4 and #5**
>
> > **Question #4** Can you add confidence intervals for all main metrics and broaden the pseudo-label noise analysis beyond the four datasets with a simple noise control, like co-teaching or small disagreement filtering?
>
> Yes, we can. To provide a robust estimate of the variability in our results across datasets, we compute bootstrap confidence intervals for the main evaluation metrics. Specifically, for each metric, we perform 1,000 bootstrap resamples over the dataset-level scores. In each iteration, we randomly sample with replacement from the scores and compute the mean of the resampled set. The 95% confidence interval is then obtained by taking the 2.5th and 97.5th percentiles of the bootstrapped mean values. This procedure ensures that the reported mean performance is accompanied by a statistically meaningful measure of uncertainty, reflecting the variability across datasets.
> The detailed results are shown in Table 5.
> ```
> Table 5. Confidence Intervals of Main Metrics for Each Method.
> ```
> | Metric/Method | Iforest | LOF | OCSVM | DeepSVDD | ECOD | GOAD | NeuTralAD | ICL | DIF | SLAD | LUNAR | MCM | DRL | TCAD |
> | --- | --- | --- | --- | --- | --- | --- | --- | --- | --- | --- | --- | --- | --- | --- |
> | auc-roc | [0.5847, 0.7246] | [0.5937, 0.6754] | [0.5433, 0.6641] | [0.5490, 0.6554] | [0.6055, 0.7252] | [0.6342, 0.8030] | [0.6010, 0.7583] | [0.6351, 0.7982] | [0.7096, 0.8774] | [0.6661, 0.8171] | [0.6038, 0.7408] | [0.7277, 0.8664] | [0.6255, 0.7874] | [0.7685, 0.9111] |
> | auc-pr | [0.4561, 0.6489] | [0.5237, 0.6524] | [0.5289, 0.6632] | [0.4153, 0.6242] | [0.4152, 0.6204] | [0.3420, 0.6320] | [0.2711, 0.5722] | [0.3415, 0.6304] | [0.4317, 0.6971] | [0.3352, 0.6624] | [0.5487, 0.6790] | [0.4138, 0.7196] | [0.2834, 0.6004] | [0.5050, 0.7717] |
> | f1 | [0.2822, 0.5360] | [0.2525, 0.4877] | [0.2578, 0.4967] | [0.2216, 0.4703] | [0.3267, 0.5329] | [0.3093, 0.5948] | [0.2425, 0.5514] | [0.3212, 0.6025] | [0.4003, 0.6499] | [0.2997, 0.6133] | [0.2585, 0.5164] | [0.3290, 0.6384] | [0.2465, 0.5570] | [0.4708, 0.7149] |
>
> Additionally, we introduce a co-teaching mechanism, where two simple MLP models teach each other by selecting low-loss subsets of samples to filter out noise in pseudo-labels. We set the forget rate to 10%, 20%, 30%, and 40%, and conducted experiments using the filtered high-confidence pseudo-labels, with results shown in Table 6. We observe that filtering more noise does not always lead to better model performance, which may be because the additional supervision from the co-teaching models can also introduce errors, causing some pseudo-labels to be incorrectly judged. Nevertheless, when the forget rate is set to 40%, the model achieves higher average performance across the 15 datasets compared to existing methods, indicating that reducing noise in pseudo-labels can further improve model performance. Therefore, in future work, developing strategies to stably enhance the quality of pseudo-labels remains an important direction.
> ```
> Table 6. Average Detection Performance Across 15 Datasets Under Different Forget Rates for Filtering Pseudo-Label Noise.
> ```
> | Forget rate/Metric | auc-roc | auc-pr | f1 |
> | --- | --- | --- | --- |
> | 0% | 0.8408 | 0.6388 | 0.5953 |
> | 10% | 0.7710 | 0.5829 | 0.5351 |
> | 20% | 0.8201 | 0.6296 | 0.5729 |
> | 30% | 0.7774 | 0.5875 | 0.5318 |
> | 40% | 0.8290 | 0.6630 | 0.6221 |
>
> > **Question #5** Can you present an appropriate cost comparison that includes end-to-end latency per test sample through all adaptation rounds, and discuss memory growth of the neighbor pool as it accumulates known normals?
>
> Yes, we can. Taking the largest dataset, Mammography, as an example, its test set contains 4,167 samples. The total inference time is 61 seconds, resulting in an average inference latency of approximately 0.015 seconds per sample. In addition, we monitored the memory usage of the normal sample pool during the adaptation process, and the results are reported in Table 7. The memory footprint of the normal pool gradually increased from 72.59 MiB at the beginning to 105.17 MiB.
> ```
> Table 7. Memory Overhead of the Normal Sample Pool During Model Adaptation.
> ```
> | Iteration | iter 1 | iter 2 | iter 3 | iter 4 | iter 5 | iter 6 | iter 7 | iter 8 | iter 9 |
> | --- | --- | --- | --- | --- | --- | --- | --- | --- | --- |
> | Memory | 72.59 | 76.66 | 80.73 | 84.80 | 88.88 | 92.95 | 97.02 | 101.09 | 105.17 |
>
> Based on these observations, we estimate that TCAD would require roughly 1 GB of memory to store the normal sample pool when handling a dataset containing approximately 100,000 samples in total. While this memory cost is sufficient for common dataset sizes, TCAD may incur relatively high memory consumption when applied to extremely large-scale datasets.
> In future work, we plan to explore strategies to reduce the size of the normal sample pool by retaining only the most representative samples. This would lower the memory consumption of TCAD when applied to extremely large-scale datasets.

---

### Official Review · Reviewer_yiGT · 2025-10-31

**Soundness:** 3
**Presentation:** 3
**Contribution:** 3
**Rating:** 6
**Confidence:** 4

**Summary:**

TCAD offers a novel and effective strategy for handling distribution shifts in unsupervised tabular anomaly detection. By integrating dual-task training with test-time contrastive learning, it enhances model robustness and sets a new state-of-the-art benchmark.

**Strengths:**

1.Tackles distribution shift between training and test normal samples—a common but overlooked issue in unsupervised tabular anomaly detection.

2.Proposes TCAD, a test-time contrastive learning framework that safely adapts to pseudo-normal samples while repelling pseudo-anomalies. Outperforms SOTA baselines on 15 datasets with constructed distribution shifts .

**Weaknesses:**

1.Method relies on masked feature reconstruction, limiting applicability to images or time series.

2.Requires prior knowledge of test-set anomaly proportion, which may not be available in practice.

3.Test-time model updates increase latency vs. static inference in standard UAD methods.

**Questions:**

1.Could TCAD be extended to image or multimodal data by replacing the masked autoencoder with a vision foundation model?

2.How sensitive is performance to misspecification of α (e.g., true α=5% but set to 20%)? Is there a way to estimate α adaptively?

3.Have you considered integrating pretrained tabular LLMs to improve initial normal pattern modeling?

4.Is the method suitable for online/streaming detection, where test samples arrive sequentially?

---

> ### Author Response · Authors · 2025-12-02
> **Responses to Weakness #1 and Question #1**
>
> > **Weakness #1 and Question #1**
> - Weakness #1: Method relies on masked feature reconstruction, limiting applicability to images or time series.
> - Question #1: Could TCAD be extended to image or multimodal data by replacing the masked autoencoder with a vision foundation model?
>
> Thank you for your comments. Regarding Weakness #1, our work is specifically designed for unsupervised anomaly detection on tabular data. Its underlying architecture is naturally better suited for tabular features and is not directly applicable to modalities such as images or time series, which possess strong spatial or sequential structures. Therefore, TCAD cannot be directly applied to image or time series data.
>
> Regarding Question #1, although the TCAD is specifically designed for unsupervised anomaly detection on tabular data and cannot be directly applied to image or multimodal data, the underlying design principles of TCAD can still serve as a reference for other modalities.
> It's noted that different modalities have distinct feature structures, types of distribution shifts, and definitions of anomalies.
> Therefore, simply replacing the masked autoencoder with a vision foundation model may not yield optimal detection performance.
> A detailed methodological design is required to account for the anomaly definitions and data characteristics of the target modality.
> Finally, while such cross-modal extensions are an interesting direction for future work, they are beyond the primary focus of the current paper.

---

> ### Author Response · Authors · 2025-12-02
> **Responses to Weakness #2 and Question #2**
>
> > **Weakness #2 and Question #2**
> - Weakness #2: Requires prior knowledge of test-set anomaly proportion, which may not be available in practice.
> - Question #2: How sensitive is performance to misspecification of $\alpha$ (e.g., true $\alpha$=5% but set to 20%)? Is there a way to estimate $\alpha$ adaptively?
>
> Thank you for your insightful comments.
> Regarding Weakness #2, we would like to clarify that our model does not use the prior contamination rate during either training or test-time adaptation. The contamination rate is only used when computing the F1-score for evaluation purposes, which is consistent with prior works [ref1, ref2].
> In addition, several studies [ref2, ref3] report only AUC-ROC or AUC-PR and therefore do not rely on the contamination rate when evaluating performance.\
> Even so, we fully acknowledge the reviewer’s concern that the true contamination rate is typically unavailable in real-world scenarios. To address this, we additionally provide a variant of our method that estimates the contamination rate in an adaptive manner. The details of this variant are provided in our response to Question #2.
>
>
> Regarding Question #2, we first set the test-set contamination rate to 5%, 10%, 15%, and 20%, respectively, to examine how the model performance changes under different mismisspecified contamination levels. The detailed results are presented in Table 1. Based on these results, we make the following observations:
> 1. Using an inaccurate contamination rate degrades the model’s detection performance.
> Compared with TCAD using the correct contamination rate, all variants that rely on incorrect contamination estimates exhibit noticeable drops in overall performance.
> 2. Metrics that depend on the contamination rate (F1-score) are affected the most, whereas AUC-ROC and AUC-PR remain relatively stable.
> ```
> Table 1. The average performance of the model across 15 datasets under different contamination rates.
> ```
> | Variants/Metric | auc-roc | auc-pr | f1 |
> | --- | --- | --- |  --- |
> | 5% | 0.7681 | 0.6127 | 0.2949 |
> | 10% | 0.7700 | 0.6148 | 0.3619 |
> | 15% | 0.7687 | 0.6050 | 0.3753 |
> | 20% | 0.7645 | 0.6051 | 0.3786 |
>
> Next, we explore a variant of our method that performs adaptive contamination-rate estimation. Estimating the contamination rate from a fully unlabeled dataset is itself a challenging standalone research problem, and there exist dedicated studies focusing on this task.
> Following this line of work, we adopt the strategy used in $\gamma$GMM [ref 5], and we define the contamination rate as: $r = \gamma GMM(D)\quad if\quad \gamma GMM(D) != 0\quad else\quad 10\%$.
> Here, $D$ denotes the dataset on which the contamination rate is estimated.
> In practice, if $\gamma$GMM is able to produce a valid contamination estimate, we directly use its output; otherwise, we default to a 10% contamination rate.
> The estimated contamination rates for all datasets are provided in Table 2.
> ```
> Table 2. The true contamination rates and the adaptively estimated contamination rates for the 15 datasets.
> ```
> | Contamination rate/Dataset |  Arrhythmia | BreastW | Cardio | Cardiotocography | Glass | Ionosphere | Mammography | Optdigits | Pendigits | Pima | Satellite | Satimage-2 | Thyroid | Wbc | Wine |
> | --- | --- | --- |  --- |  --- |  --- |  --- |  --- |  --- |  --- |  --- |  --- |  --- |  --- |  --- |  --- |
> | Ture | 0.2727 | 0.6176 | 0.1681 | 0.3923 | 0.1071 | 0.5806 | 0.0624 | 0.058 | 0.0461 | 0.5595 | 0.4776 | 0.0237 | 0.052 | 0.1489 | 0.1471 |
> | Estimated | 0.1 | 0.1 | 0.1002 | 0.0215 | 0.085 | 0.1 | 0.0702 | 0.0992 | 0.0305 | 0.0562 | 0.1061 | 0.1 | 0.0463 | 0.1 | 0.1048 |
>
> As shown in Table 2, we observe that the estimated contamination rates differ substantially from the true contamination rates.
> To further verify that our method remains competitive under such estimation errors, we compare TCAD with MCM, which also requires the contamination rate for computing the F1-score. The results are presented in Table 3. TCAD achieves higher AUC-PR and F1-scores, demonstrating its strong capability in detecting anomalous samples even when the contamination rate is inaccurately estimated.
> ```
> Table A3. Comparison of TCAD and MCM on 15 Datasets Using Estimated Contamination Rates.
> ```
> | Mehtod/Metric | auc-roc | auc-pr | f1 |
> | --- | --- | --- |  --- |
> | MCM | 0.8136	| 0.5957 | 0.3283 |
> | TCAD | 0.7398 | 0.6158 | 0.3702 |
>
>
> -[ref 1] MCM: Masked cell modeling for anomaly detection in tabular data, ICLR 2024.\
> -[ref 2] Anomaly detection for tabular data with internal contrastive learning, ICLR 2022.\
> -[ref 3] Fascinating supervisory signals and where to find them: Deep anomaly detection with scale learning, ICML 2023.\
> -[ref 4] ECOD: Unsupervised outlier detection using empirical cumulative distribution functions, TKDE 2022. \
> -[ref 5] Estimating the contamination factor’s distribution in unsupervised anomaly detection, ICML 2023.

---

> ### Author Response · Authors · 2025-12-02
> **Responses to Weakness #3, Questions #3 and #4**
>
> > **Weakness #3** Test-time model updates increase latency vs. static inference in standard UAD methods.
>
> Thank you for your comment. We acknowledge that performing model updates at test time introduces additional computational overhead. However, this procedure also brings performance improvements. Using MCM as a baseline, we measured the average inference time and memory usage of both MCM and TCAD across 15 datasets (see Table 4), which shows that TCAD incurs slightly higher time and memory costs.
> At the same time, according to the main results reported in our paper, TCAD achieves an average F1-score improvement of over 0.1 compared to MCM (from 0.4870 to 0.5953), demonstrating the effectiveness of trading a modest increase in computational cost for improved detection performance.
> ```
> Table 4. Comparison of Memory Overhead (MiB) and Runtime (Second).
> ```
> | Methods | Average overhead across 15 datasets |
> | --- | --- |
> | MCM-MEM | ~1865 |
> | TCAD-MEM | ~2121 |
> | MCM-TIME | ~86 |
> | TCAD-TIME | ~187 |
>
> Furthermore, we conducted a detailed analysis of the main source of the increased computational cost in TCAD. The KNN-based embedding query used in our current implementation is performed on the CPU, and its runtime increases noticeably as the dataset size grows. To address this, we plan to replace the current KNN query with a GPU-based implementation in future code updates, which is expected to significantly reduce the model’s runtime.
>
> > **Question #3** Have you considered integrating pretrained tabular LLMs to improve initial normal pattern modeling?
>
> Thank you for your suggestion. In the current work, we have not explored leveraging pretrained tabular LLMs for modeling normal samples. While pretrained models may provide improved data representation, our method framework still requires a small amount of training or optimization during test-time adaptation. Performing full training or fine-tuning of large pretrained models would incur substantial computational costs. In our recent work, we are exploring the use of LLMs as backbone models for zero-shot and few-shot anomaly detection, and we expect to provide a detailed report on these investigations in future work.
>
> > **Question #4** Is the method suitable for online/streaming detection, where test samples arrive sequentially?
>
> Thank you for your question. Our method can process test data in a batch-wise manner, which allows for a detection mode somewhat similar to online/streaming scenarios. However, TCAD is not fundamentally designed for true online or streaming detection. Therefore, careful design is still needed to enable faster processing of individual samples and effective updating and maintenance of incremental models. This represents an important direction for future work, and we look forward to exploring it in our subsequent research.

---

### Official Review · Reviewer_fH6Y · 2025-11-03

**Soundness:** 2
**Presentation:** 2
**Contribution:** 1
**Rating:** 2
**Confidence:** 3

**Summary:**

This paper proposes TCAD, an approach for unsupervised anomaly detection in tabular data designed to address distribution shifts where normal patterns in the test set differ from those in the training set. TCAD operates in two  stages: Collaborative Dual-task Training and Test-Time Contrastive Learning. During training, it uses two self-supervised tasks to capture features and model normal patterns. At test time, the model adapts by assigning pseudo-labels (normal or abnormal) to high-confidence samples. It then updates to adapt to these pseudo-normal samples while avoiding overfitting to pseudo-abnormal ones. A KNN-based contrastive strategy then pulls pseudo-normal samples toward the training distribution’s embeddings and pushes pseudo-abnormal samples away.

**Strengths:**

Designing robust anomaly detectors that generalize well to new domains is critical.

The paper states its goals and contributions clearly.

**Weaknesses:**

W1) Anomaly detection under distribution shift has been explored in computer vision [A,B,C]; it is unclear why the authors did not cite or discuss this literature.


W2) The pipeline’s technical novelty is the main issue. Contrastive loss and reconstruction loss are well known and widely used in the literature. Selecting samples with high confidence at test time is also a known technique [D]. Can the authors describe the components that genuinely belong to their method?

W3) I believe anomaly detection under distribution shift is better defined in the vision domain, as foreground and background in images provide a well-defined approach for specifying shifted normal or abnormal data. The authors should evaluate their pipeline on those datasets as well.


W4) The code is not available, making it challenging to reproduce the results.


[A] Robust Novelty Detection through Style-Conscious Feature Ranking

[B] A Contrastive Teacher-Student Framework for Novelty Detection under Style Shifts

[C] Red PANDA: Disambiguating Anomaly Detection by Removing Nuisance Factors

[D] SCAN: Learning to Classify Images without Labels

**Questions:**

See Weaknesses.

---

> ### Author Response · Authors · 2025-12-02
> **Response to Weakness #1 and #2**
>
> >**Weakness #1**
>  Anomaly detection under distribution shift has been explored in computer vision [A,B,C]; it is unclear why the authors did not cite or discuss this literature.
>
> Thank you for your comment. We would like to clarify that our work focuses on unsupervised anomaly detection in tabular data, rather than tasks in the computer vision (CV) domain, as reflected in the title of our paper, “Unsupervised Anomaly Detection in Tabular Data with Test-time Contrastive Learning.”
>
> There are several key distinctions between our work and existing CV-based methods:
> 1. **Data structure differences:**
> Images exhibit strong spatial structures, whereas tabular data consist of independent features without inherent spatial correlations. As a result, many CV methods (e.g., shift modeling based on convolutional architectures) cannot be directly applied to tabular data.
> 2. **Types of distribution shifts:**
> Distribution shifts in CV typically involve changes in texture, illumination, background, or viewpoint. In contrast, distribution shifts in tabular data commonly manifest as feature distribution drift or changes in statistical properties, which require different modeling strategies.
> 3. **Definition of anomalies:**
> In CV domain, anomalies often correspond to semantic deviations at the object or region level. For tabular data, anomalies are high-dimensional statistical deviations or abstract abnormal patterns, necessitating fundamentally different detection mechanisms.
>
> Given these substantial differences, our literature review and experimental comparisons focus on work within the tabular data domain, and CV-based methods were not included in the main discussion.
>
>
> >**Weakness #2**
> The pipeline’s technical novelty is the main issue. Contrastive loss and reconstruction loss are well-known and widely used in the literature. Selecting samples with high confidence at test time is also a known technique [D]. Can the authors describe the components that genuinely belong to their method?
>
> Thank you for your comment.
> We fully agree that contrastive loss, reconstruction loss, and high-confidence sample selection are well-established concepts in the literature. **However, the core focus of our paper is not the novelty of these components individually, but rather how to design an effective framework for unsupervised anomaly detection on tabular data under distribution shift.**
>
> More concretely, our method introduces the following key innovations, which are unique to our approach:
> 1. **A Collaborative Dual-task Training scheme tailored for tabular data:**
> We design two complementary self-supervised masked reconstruction tasks: one operates at the feature level and the other at the semantic level.
> They jointly model normal patterns in tabular data.
> This dual-task design allows the model to capture both low-level feature representations and high-level semantic knowledge, which is crucial for learning robust representations of normal behavior in tabular domains.
>
> 2. **Test-Time Contrastive Learning guided by both pseudo-normal and pseudo-abnormal labels:**
> Prior test-time training approaches either directly adapt the model to test data or restrict adaptation to presumed normal samples.
> These methods typically avoid adapting to anomalous samples due to the risk of corrupting the model’s ability to detect anomalies.
> In contrast, our method allows the model to cautiously utilize both pseudo-normal and pseudo-abnormal samples during test-time adaptation.
> To effectively leverage both pseudo-normal and pseudo-abnormal samples while preventing the model from degrading its detection capability, we introduce a K-nearest-neighbor contrastive learning strategy that constrains their relative positions in the embedding space. Specifically, pseudo-normal samples are encouraged to move closer to similar known normal samples, whereas pseudo-abnormal samples are pushed away from nearby normal instances. This design enables the model to maintain clear separation between normal and abnormal patterns in the embedding space, thereby avoiding performance deterioration during the adaptation process.
>
> In summary, our method is not a simple aggregation of well-known components. Instead, it introduces a novel and coherent framework specifically designed for unsupervised anomaly detection on tabular data under distribution shift. The components are tightly coupled with the task objectives, and their interactions, optimization dynamics, and adaptation procedures differ fundamentally from existing approaches.

---

> ### Author Response · Authors · 2025-12-02
> **Responses to Weakness #3 and #4**
>
> > **Weakness #3** I believe anomaly detection under distribution shift is better defined in the vision domain, as foreground and background in images provide a well-defined approach for specifying shifted normal or abnormal data. The authors should evaluate their pipeline on those datasets as well.
>
> Thank you for your comment.
> We agree that in the vision domain, the definition of distribution shift is more intuitive due to the clear semantic separation between foreground and background.
> However, as we mentioned in our response to Weakness #1, tabular data differ fundamentally from visual data in terms of data structure, the types of distribution shifts that occur, and the way anomalies are defined. These differences lead to research objectives and focuses that are not aligned with methods developed for computer vision.
>
> To address the unique challenges in tabular anomaly detection, our proposed framework is specifically designed around the characteristics of tabular data. This includes the representation learning strategy, the design of the self-supervised tasks, and the embedding adaptation mechanism used during test time. Owing to these modality-specific considerations, our method differs substantially from approaches developed for the CV setting.
>
> In summary, our method differs substantially from vision-based approaches in terms of research objectives, design motivations, and methodological details. For these reasons, evaluating our pipeline on visual datasets would not provide a fair or meaningful assessment of its effectiveness.
>
> Nonetheless, we appreciate the reviewer’s insight. As part of our future work, we plan to explore distribution shift problems in the vision domain as well as extend our framework to multimodal settings.
>
>
> > **Weakness #4** The code is not available, making it challenging to reproduce the results.
>
> Thank you for your comment. We highly value the reproducibility of our work and have clearly stated in the appendix that the full source code will be released after the anonymity period. In fact, the complete codebase has already been fully organized and uploaded to GitHub.
> However, to prevent potential identity disclosure and to protect our code contributions, the repository is currently set to private.
> Once the review process concludes and we receive the final notification, we will immediately switch the repository to public, enabling the community to fully reproduce and verify our results.

---

### Meta-Review · Area_Chair_bdUZ · 2025-12-07

**Summary:**

Two out of three reviewers are prone to reject the paper.  Authors present the paper from a general perspective at the beginning of the abstract and introduction sections, that is the reason why  reviewer raised the concern about distribution shift and anomaly detection in computer vision (CV). The reviewer concern about using known components (contrastive loss, reconstruction loss, and high-confidence sample selection), which should be respected. Reviewers hope to see the improvement of the well-known loss function for Tabular Data but the paper fail to do.

**Reviewer Concerns:**

fH6Y about distribution shift and anomaly detection should be addressed in computer vision (CV). The reviewer jpNR concern about using known components (contrastive loss, reconstruction loss, and high-confidence sample selection) should be respected.

**Reviewer Scores:**

Two out of three reviewers are negative on the key contribution of the paper with score of 2 and 4.
Another reviewer still suggest experiment on a vision foundation model, but authors seems to insist on their claim on a specific application.

---

### Decision · Program_Chairs · 2026-01-26

Reject